# $t^3$-Variational Autoencoder: Learning Heavy-tailed Data with Student's t and Power Divergence

**Juno Kim[1,2]*   Jaehyuk Kwon[3]*   Mincheol Cho[3]*   Hyunjong Lee[3]   Joong-Ho Won[3]**

[1]Department of Mathematical Informatics, The University of Tokyo
[2]Center for Advanced Intelligence Project, RIKEN
[3]Department of Statistics, Seoul National University

junokim@g.ecc.u-tokyo.ac.jp  {jh19984,code1478,hyunjong526}@snu.ac.kr  wonj@stats.snu.ac.kr

## ABSTRACT

The variational autoencoder (VAE) typically employs a standard normal prior as a regularizer for the probabilistic latent encoder. However, the Gaussian tail often decays too quickly to effectively accommodate the encoded points, failing to preserve crucial structures hidden in the data. In this paper, we explore the use of heavy-tailed models to combat over-regularization. Drawing upon insights from information geometry, we propose $t^3$VAE, a modified VAE framework that incorporates Student's t-distributions for the prior, encoder, and decoder. This results in a joint model distribution of a power form which we argue can better fit real-world datasets. We derive a new objective by reformulating the evidence lower bound as joint optimization of KL divergence between two statistical manifolds and replacing with $\gamma$-power divergence, a natural alternative for power families. $t^3$VAE demonstrates superior generation of low-density regions when trained on heavy-tailed synthetic data. Furthermore, we show that $t^3$VAE significantly outperforms other models on CelebA and imbalanced CIFAR-100 datasets.

## 1 INTRODUCTION

The variational autoencoder (VAE, Kingma & Welling, 2013) is a popular probabilistic generative model for learning compact latent data representations. The VAE consists of two conditional models: an encoder which models the posterior distribution of the latent variable $z$ given an observation $x$, and a decoder which infers the observation from its latent representation, which are jointly trained by optimizing the evidence lower bound (ELBO). The VAE framework *a priori* does not require the prior, encoder or decoder to be a particular probability distribution; the usual choice of Gaussian is mainly due to feasibility of the reparametrization trick and closed-form computation of divergence.

However, real-world data frequently exhibits outlier-heavy or heavy-tailed behavior which is better captured by models of similar nature. Recently, Floto et al. (2023) showed that the Gaussian VAE encodes many points in low-density regions of the prior; the distribution is too tight to effectively fit complex latent representations. We argue that distributing more mass to the tails allows encoded points to spread out easily, leading us to adopt a Student's t-distributed prior. We also incorporate a t-distributed decoder which amplifies variability of data generated from low-density regions. From a Bayesian perspective, this is equivalent to incorporating a latent precision affecting both $x, z$. Together with the prior, this results in a joint model $p_{\theta,\nu}(x, z)$ of a power form, generalizing the exponential form of the original VAE analogously to how the t-distribution generalizes the Gaussian.

Changing the distributions usually necessitates numerical integration to estimate the ELBO. We provide a novel alternative based on recent theoretical insights from information geometry. Han et al. (2020) showed that the ELBO can be reformulated as minimization of KL divergence between two statistical manifolds. Separately, Eguchi (2021) has developed a theory of $\gamma$-power divergence that parallels KL divergence. In this new geometry, power families play the role of exponential families,

---

*Equal contribution.

providing a natural setting for joint optimization of heavy-tailed models. By minimizing $\gamma$-power instead of KL divergence, we construct a general-purpose framework implementing t-distributions for the prior, encoder and decoder and a new objective called $\gamma$-loss, which we call the $t^3$VAE.

$t^3$VAE requires a single hyperparameter $\nu$ which is coupled to the degrees of freedom of the t-distributions and controls as a balance between reconstruction and regularization. In particular, $t^3$VAE encompasses the Gaussian VAE and ordinary autoencoder as the limiting cases $\nu \to \infty$ and $\nu \to 2$, respectively. The $\gamma$-loss has an approximate closed form analogous to the ELBO and can be optimized via a t-reparametrization trick. We empirically demonstrate that $t^3$VAE can successfully approximate low-density regions of heavy-tailed datasets. Furthermore, $t^3$VAE is able to learn and generate high-dimensional images in richer detail compared to various alternative VAEs. Finally, we extend our model to a hierarchical architecture, the $t^3$HVAE, which is able to reconstruct high-resolution images with more sophistication.

## 1.1 RELATED WORKS

Many authors point out the standard normal prior can induce over-regularization, losing valuable semantics hidden in the data. Alternative priors based on Gaussian mixtures (Tomczak & Welling, 2018; Dilokthanakul et al., 2016), the Dirichlet distribution (Joo et al., 2020), the von Mises-Fisher distribution (Davidson et al., 2018), normalizing flows (Rezende & Mohamed, 2015) or stochastic processes (Goyal et al., 2017; Nalisnick & Smyth, 2017) have been proposed to mitigate this influence.

Another line of research has focused on modifying the ELBO using different divergences (Li & Turner, 2016; Deasy et al., 2021) or weights (Higgins et al., 2017). Instead of only changing the KL regularizer, we take advantage of the joint KL divergence formulation for the first time. The $\gamma$-power and similar divergences have been studied before in the robust statistics literature. Some examples are density power divergence (Basu et al., 1998), logarithmic $\gamma$-divergence (Fujisawa & Eguchi, 2008), and robust variational inference (Futami et al., 2018).

Some previous works have studied VAEs incorporating the t-distribution, namely the Student-$t$ VAE with a t-decoder (Takahashi et al., 2018), the VAE-st with a t-prior and t-encoder (Abiri & Ohlsson, 2020), and the 'disentangled' or DE-VAE with a product of t-priors (Mathieu et al., 2019). Unlike $t^3$VAE, none of these stray from the ELBO framework, and the latter two require numerical integration to compute the KL regularizer. We also point out that these models all use a product of univariate t-distributions while $t^3$VAE uses the multivariate t-distribution. We implement and compare with these models in our experiments.

Finally, heavy-tailed distributions have also been used as base densities in the normalizing flow literature (Alexanderson & Henter, 2020; Jaini et al., 2020; Laszkiewicz et al., 2022; Amiri et al., 2022), where it has been argued that t-distributions lead to improved robustness and generalization. We take a step further by enforcing a power form on the joint model, which is key to $t^3$VAE's success.

## 2 THEORETICAL BACKGROUND

In this section, we summarize key aspects of the motivating theories of variational inference and information geometry. Details and proofs are deferred to Appendix A.

## 2.1 VAE AS JOINT MINIMIZATION

Formally, a VAE models the distribution $p_{\text{data}}(x)$ of the observed variable $x \in \mathbb{R}^n$ by jointly learning a stochastic latent variable $z \in \mathbb{R}^m$. Generation is performed by sampling $z$ from the prior $p_Z(z)$, then sampling $x$ according to a probabilistic decoder $p_\theta(x|z)$ parametrized by $\theta \in \Theta$. The observed likelihood $p_\theta(x) = \int p_\theta(x|z) p_Z(z) dz$ is intractable, so we instead aim to approximate the posterior $p_\theta(z|x)$ with a parametrized encoder $q_\phi(z|x)$ by minimizing their KL divergence. This leads to maximizing the *evidence lower bound* (ELBO) of the log-likelihood, defined as

$$\mathcal{L}(x; \theta, \phi) := \log p_\theta(x) - \mathcal{D}_{\text{KL}}(q_\phi(z|x) \,\|\, p_\theta(z|x)) \tag{1}$$

$$= \mathbb{E}_{z \sim q_\phi(\cdot|x)} [\log p_\theta(x|z)] - \mathcal{D}_{\text{KL}}(q_\phi(z|x) \,\|\, p_Z(z)). \tag{2}$$

Since $-\mathbb{E}_{z \sim q_\phi(\cdot|x)} [\log p_\theta(x|z)]$ is the cross-entropy reconstruction loss of the original data, the ELBO can be understood as adding a KL regularization term, forcing $q_\phi(z|x)$ close to the prior

in order to ensure that the latent state $z$ encodes only useful information. In practice, the prior is typically standard normal and the encoder and decoder distributions are parametrized Gaussians,

$$p_Z(z) \sim \mathcal{N}_m(0, I), \quad q_\phi(z|x) \sim \mathcal{N}_m(\mu_\phi(x), \Sigma_\phi(x)), \quad p_\theta(x|z) \sim \mathcal{N}_n(\mu_\theta(z), \sigma^2 I) \quad (3)$$

in which case the reconstruction loss is equal to the mean squared error (MSE) between $x$ and the decoded mean up to a constant,

$$\mathbb{E}_{z \sim q_\phi(\cdot|x)} [\log p_\theta(x|z)] = \mathbb{E}_{z \sim q_\phi(\cdot|x)} \left[ -\frac{1}{2\sigma^2} \|x - \mu_\theta(z)\|^2 \right] + \text{const.} \quad (4)$$

The encoder covariance is usually taken as (but not assumed to be) diagonal, $\Sigma_\phi(x) = \text{diag}\, \sigma_\phi^2(x)$.

Han et al. (2020) point out that the VAE can be reinterpreted as a joint minimization process between two statistical manifolds. Let $\mathcal{P} = \{p_\theta(x, z) = p_\theta(x|z)p_Z(z) : \theta \in \Theta\}$ be the *model distribution manifold*, and $\mathcal{Q} := \{q_\phi(x, z) = p_{\text{data}}(x)q_\phi(z|x) : \phi \in \Phi\}$ the *data distribution manifold*, both finite-dimensional submanifolds of the space of joint distributions. The divergence between points on $\mathcal{P}, \mathcal{Q}$ may be recast (Appendix A.1) as

$$\mathcal{D}_{\text{KL}}(q_\phi(x, z) \,\|\, p_\theta(x, z)) = -\mathbb{E}_{x \sim p_{\text{data}}} [\mathcal{L}(x; \theta, \phi)] - H(p_{\text{data}}), \quad (5)$$

where $H(p_{\text{data}})$ denotes the differential entropy of the data distribution. Thus, maximizing the ELBO is equivalent to finding the joint minimizer

$$(p_{\theta^*}, q_{\phi^*}) = \underset{p \in \mathcal{P},\, q \in \mathcal{Q}}{\text{argmin}}\ \mathcal{D}_{\text{KL}}(q \,\|\, p). \quad (6)$$

## 2.2 INFORMATION GEOMETRY AND $\gamma$-POWER DIVERGENCE

Any divergence $\mathcal{D}$ on a statistical manifold $\mathcal{M}$ of probability distributions automatically induces a Riemannian metric $g$ and two affine connections $\Gamma, \Gamma^*$ on $\mathcal{M}$ dually coupled with respect to $g$ (Amari, 2016, Chapter 6). For KL divergence, $g$ is the Fisher information metric and the $\Gamma$ and $\Gamma^*$-autoparallel curves connecting two points $p, q$ are the $m$-(mixture) geodesic $p_t(x) = (1 - t)p(x) + tq(x)$ and the $e$-(exponential) geodesic $p_t(x) \propto \exp((1 - t)\log p(x) + t \log q(x))$, respectively. In particular, the $\Gamma^*$-flat submanifolds consist of the exponential families. The problem (6) may then be solved by applying the information geometric $em$-algorithm (Csiszár, 1984; Han et al., 2020).

Paralleling the KL case, Eguchi (2021) defines the $\gamma$-power entropy and cross-entropy for probability distributions $q, p$ with power $\gamma$ as

$$\mathcal{H}_\gamma(p) := -\|p\|_{1+\gamma} = -\left( \int p(x)^{1+\gamma} dx \right)^{\frac{1}{1+\gamma}}, \quad \mathcal{C}_\gamma(q, p) := -\int q(x) \left( \frac{p(x)}{\|p\|_{1+\gamma}} \right)^\gamma dx \quad (7)$$

and the $\gamma$-power divergence as $\mathcal{D}_\gamma(q \,\|\, p) = \mathcal{C}_\gamma(q, p) - \mathcal{H}_\gamma(q)$. In our paper, we introduce an additional factor of $1/\gamma$ in order to extend to the case $-1 < \gamma < 0$,[1]

$$\mathcal{D}_\gamma(q \,\|\, p) := \gamma^{-1} \mathcal{C}_\gamma(q, p) - \gamma^{-1} \mathcal{H}_\gamma(q). \quad (8)$$

We show that $\mathcal{D}_\gamma$ is indeed a divergence and derive its induced metric in Appendix A.2. Computing the dual connections yields that the totally $\Gamma^*$-geodesic (or $\gamma$-flat) submanifolds consist of power families of the form

$$\mathcal{S}_\gamma = \{p_\theta(x) \propto (1 + \gamma\theta^\top s(x))^{\frac{1}{\gamma}} : \theta \in \Theta\}. \quad (9)$$

In particular, the family of $d$-variate Student's t-distributions with variable mean $\mu$, scale matrix $\Sigma$ and fixed degrees of freedom $\nu$

$$t_p(x|\mu, \Sigma, \nu) = C_{\nu,d} |\Sigma|^{-\frac{1}{2}} \left( 1 + \frac{1}{\nu}(x - \mu)^\top \Sigma^{-1}(x - \mu) \right)^{-\frac{\nu+d}{2}}, \quad C_{\nu,d} = \frac{\Gamma(\frac{\nu+d}{2})}{\Gamma(\frac{\nu}{2})(\nu\pi)^{\frac{d}{2}}} \quad (10)$$

is $\gamma$-flat when $\gamma = -\frac{2}{\nu+d}$, which we assume in the remainder of this Section. Furthermore, the $\gamma$-power divergence from $q_\nu \sim t_d(\mu_0, \Sigma_0, \nu)$ to $p_\nu \sim t_d(\mu_1, \Sigma_1, \nu)$ is finite when $\nu > 2$ and can be

---

[1]Eguchi (2021) also points out that the t-distribution naturally emerges as the maximal entropy distribution when $\gamma = -\frac{2}{\nu+1}$, see Proposition 2. However, his original definition of $\gamma$-power divergence is erroneously non-negative only for $\gamma > 0$, necessitating our modification.

expressed in closed-form (Proposition 3) as

$$
\begin{aligned}
\mathcal{D}_\gamma(q_\nu \,\|\, p_\nu) = &-\tfrac{1}{\gamma} C_{\nu,d}^{\frac{\gamma}{1+\gamma}} \left(1 + \tfrac{d}{\nu-2}\right)^{-\frac{\gamma}{1+\gamma}} \left[ -|\Sigma_0|^{-\frac{\gamma}{2(1+\gamma)}} \left(1 + \tfrac{d}{\nu-2}\right) \right. \\
&\left. + |\Sigma_1|^{-\frac{\gamma}{2(1+\gamma)}} \left(1 + \tfrac{1}{\nu-2} \operatorname{tr}\left(\Sigma_1^{-1}\Sigma_0\right) + \tfrac{1}{\nu}(\mu_0 - \mu_1)^\top \Sigma_1^{-1}(\mu_0 - \mu_1)\right) \right].
\end{aligned}
\tag{11}
$$

KL divergence may be retrieved from $\gamma$-power divergence as $\lim_{\gamma \to 0} \mathcal{D}_\gamma(q \,\|\, p) = \mathcal{D}_{\mathrm{KL}}(q \,\|\, p)$, see Proposition 4. Moreover, Equation (11) converges to the KL divergence between the limiting Gaussian distributions $q_\infty \sim \mathcal{N}_d(\mu_0, \Sigma_0)$ and $p_\infty \sim \mathcal{N}_d(\mu_1, \Sigma_1)$ as $\nu \to \infty$.

## 3 THE $t^3$-VARIATIONAL AUTOENCODER

### 3.1 STRUCTURE OF THE $t^3$VAE

Throughout this section, we present definitions and theoretical properties of our $t^3$VAE model and $\gamma$-loss function. Full derivations are provided in Appendix B. When the prior, encoder and decoder are normally distributed as in Equation (3), the joint model distribution $p_\theta(x, z) \in \mathcal{P}$ takes the form

$$
p_\theta(x, z) \propto \sigma^{-n} \exp\left[ -\frac{1}{2}\left( \|z\|^2 + \frac{1}{\sigma^2}\|x - \mu_\theta(z)\|^2 \right) \right].
\tag{12}
$$

Since the tail is exponentially bounded, its capacity to approximate the data distribution manifold corresponding to real-world data is limited. To mitigate this problem, we propose a heavy-tailed model $p_{\theta,\nu}(x, z)$ of a power form, parametrized by the degrees of freedom $\nu > 2$,

$$
p_{\theta,\nu}(x, z) \propto \sigma^{-n} \left[ 1 + \frac{1}{\nu}\left( \|z\|^2 + \frac{1}{\sigma^2}\|x - \mu_\theta(z)\|^2 \right) \right]^{-\frac{\nu+m+n}{2}},
\tag{13}
$$

which is obtained from the prior and decoder distributions

$$
p_{Z,\nu}(z) = t_m\left(z | 0, I, \nu\right),
\tag{14}
$$

$$
p_{\theta,\nu}(x|z) = t_n\left(x \,\Big|\, \mu_\theta(z), \frac{1 + \nu^{-1}\|z\|^2}{1 + \nu^{-1}m}\sigma^2 I, \nu + m\right).
\tag{15}
$$

Since the true posterior $z|x$ is t-distributed with degrees of freedom $\nu + n$ when the decoder is shallow (discussed in Appendix B.2), we are also motivated to incorporate a t-distributed encoder

$$
q_{\phi,\nu}(z|x) = t_m\left(z \,\Big|\, \mu_\phi(x), (1 + \nu^{-1}n)^{-1}\Sigma_\phi(x), \nu + n\right).
\tag{16}
$$

Hence, the $t^3$VAE model and data distribution manifolds are $\mathcal{P}_\nu = \{p_{\theta,\nu}(x, z) = p_{\theta,\nu}(x|z)p_{Z,\nu}(z) : \theta \in \Theta\}$ and $\mathcal{Q}_\nu = \{q_{\phi,\nu}(x, z) = p_{\mathrm{data}}(x)q_{\phi,\nu}(z|x) : \phi \in \Phi\}$. This generalizes the Gaussian VAE in the sense that $p_{\theta,\nu}(x, z)$ and $q_{\phi,\nu}(\cdot|x)$ uniformly converge to $p_\theta(x, z)$ and $q_\phi(\cdot|x)$ as $\nu \to \infty$.

### 3.2 $\gamma$-POWER DIVERGENCE LOSS

From the geometric relationship of $\gamma$-power divergence and power families and the model equation (13), we are motivated to replace the KL objective in the joint minimization problem (6) with $\gamma$-power divergence for $t^3$VAE,

$$
(p_{\theta^*,\nu}, q_{\phi^*,\nu}) = \operatorname*{argmin}_{p \in \mathcal{P}_\nu,\, q \in \mathcal{Q}_\nu} \mathcal{D}_\gamma(q \,\|\, p)
\tag{17}
$$

where $\gamma$ is coupled to $\nu$ as $\gamma = -\frac{2}{\nu+n+m}$. The $\gamma$-power divergence from $q_{\phi,\nu} \in \mathcal{Q}_\nu$ to $p_{\theta,\nu} \in \mathcal{P}_\nu$ can be computed in closed-form after an approximation of order $\gamma^2$ (Proposition 5). After rearranging constants, we obtain the $\gamma$-**loss** objective which is amenable to Monte Carlo estimation,

$$
\begin{aligned}
\mathcal{L}_\gamma(\theta, \phi) = \frac{1}{2}\, \mathbb{E}_{x \sim p_{\mathrm{data}}} \Bigg[ &\frac{1}{\sigma^2} \mathbb{E}_{z \sim q_{\phi,\nu}(\cdot|x)} \|x - \mu_\theta(z)\|^2 \\
&+ \|\mu_\phi(x)\|^2 + \frac{\nu}{\nu+n-2} \operatorname{tr}\Sigma_\phi(x) - \frac{\nu C_1}{C_2} |\Sigma_\phi(x)|^{-\frac{\gamma}{2(1+\gamma)}} \Bigg]
\end{aligned}
\tag{18}
$$

for constants $C_1 = \left(\frac{\nu+m+n-2}{\nu+n-2}\left(1 + \frac{n}{\nu}\right)^{\frac{\gamma m}{2}} C_{\nu+n,m}^\gamma\right)^{\frac{1}{1+\gamma}}$ and $C_2 = \left(\frac{\nu+m+n-2}{\nu-2}\sigma^n C_{\nu,m+n}^{-1}\right)^{-\frac{\gamma}{1+\gamma}}$.
See Appendix B.3 for proofs and analysis of error.

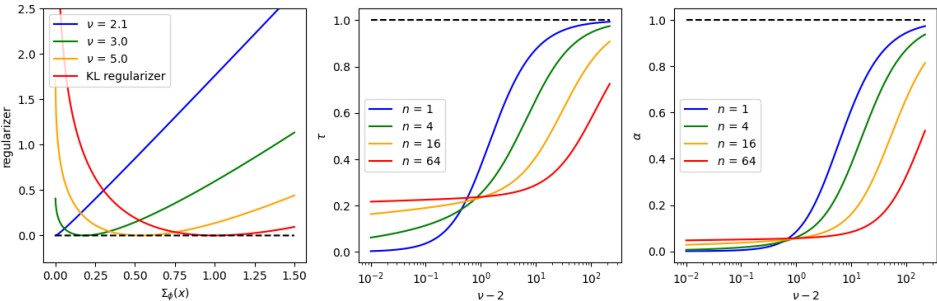

Figure 1: (a) Dependency of regularization on $\Sigma_\phi(x)$ when $m = n = 1$, $\sigma = 1$ (left); (b) graph of the alternative prior scale $\tau$ against $\nu$ (middle), (c) graph of the regularizer coefficient $\alpha$ against $\nu$ (right).

The $\gamma$-loss does not lower bound the data likelihood; its precise role will be made clear shortly. We emphasize that our framework is to view the ELBO *not* as a bound of likelihood (which leads to modifying the KL regularizer), but as a divergence between joint distributions (leads to modifying the entire divergence). This change is further justified in Fujisawa & Eguchi (2008) which shows that $\gamma$-type divergence minimizers are asymptotically efficient M-estimators, lending support to our approach as a solid alternative to maximum likelihood methods for statistical inference.

### 3.3 $\nu$ CONTROLS REGULARIZATION STRENGTH

We now shine light on the meaning of the $t^3$VAE hyperparameter $\nu$. Analogously to the ELBO, the $\gamma$-loss (18) consists of an MSE reconstruction error and additional terms which act as a regularizer. This can be made precise: in fact, the remaining terms are equivalent (up to constants) to the $\gamma$-power divergence from the posterior $q_{\phi,\nu}(z|x)$ to the *alternative* prior

$$p_\nu^\star(z) = t_m\left(z|0, \tau^2 I, \nu + n\right); \quad \tau^2 = \frac{1}{1+\nu^{-1}n}\left(\sigma^{-n}C_{\nu,n}(1 + \frac{n}{\nu-2})^{-1}\right)^{\frac{2}{\nu+n-2}}, \quad (19)$$

derived in Appendix B.4. Equation (18) can then be rewritten as

$$\mathcal{L}_\gamma(\theta, \phi) = \mathbb{E}_{x \sim p_{\text{data}}}\left[\frac{1}{2\sigma^2}\mathbb{E}_{z \sim q_{\phi,\nu}(\cdot|x)}\|x - \mu_\theta(z)\|^2 + \alpha \mathcal{D}_\gamma(q_{\phi,\nu} \| p_\nu^\star)\right] + \text{const.} \quad (20)$$

Hence, $\gamma$-loss can be interpreted similarly as a balance between reconstruction and regularization, and $\nu$ controls both the target scale $\tau^2$ and the regularizer coefficient $\alpha = -\frac{\gamma\nu}{2C_2}$.

Figure 1(a) plots the regularizer as a function of $\Sigma_\phi(x)$ for a range of $\nu$ when $m = n = 1$. The KL graph verifies that $\Sigma_\phi(x)$ is forced towards $\tau^2 = 1$ in the Gaussian case, and this effect is particularly strong when $\Sigma_\phi(x) < \tau^2$. For smaller degrees of freedom, $\tau$ becomes less than 1 and the forcing effect for small $\Sigma_\phi(x)$ decreases as well, allowing the encoded distributions to be more point-like. The overall weakening of regularization is consistent with our motivations for adopting a heavy-tailed prior. This also separates $t^3$VAE from models which simply assign a weight to the KL regularizer such as $\beta$-VAE (Higgins et al., 2017), where the target $\tau^2$ remains constant.

Figure 1(b),(c) further show the dependency of $\tau, \alpha$ against $\nu$ for $n \in \{1, 4, 16, 64\}$. As $\nu \to \infty$, $t^3$VAE converges to the Gaussian VAE. As $\nu \to 2$, in theory both $\tau, \alpha \to 0$ so that regularization vanishes and $t^3$VAE regresses to a raw autoencoder. However, this does not occur in practice in high dimensions as the model is much less sensitive to $\nu$ near 2 due to the slow convergence rate. In low dimensions, this regime does come into play and $t^3$VAE performs significantly worse if $\nu$ is very small. We therefore suggest selecting $\nu - 2$ to be roughly logarithmically spaced to observe varying levels of regularization strength, and only a few values are needed for higher dimensional datasets.

### 3.4 $t^3$HVAE: THE BAYESIAN HIERARCHY

The principle of $t^3$VAE naturally extends to hierarchical VAEs (HVAEs), which have recently shown great success in generating high-quality images and learning structured features (Vahdat & Kautz, 2020; Child, 2021; Havtorn et al., 2021). In Appendix B.1, we prove $t^3$VAE may be viewed as a

---

**Algorithm 1:** Overview of $t^3$VAE

---

**Require** data $x^{(1)}, \cdots, x^{(K)}$, hyperparameter $\nu$
compute $\gamma, \tau, C_1, C_2$
**for** $i \in \{1, \ldots, K\}$ **do**
    retrieve $\mu_\phi(x^{(i)}), \Sigma_\phi(x^{(i)})$
    sample $z^{(i)} \sim q_{\phi,\nu}(z|x^{(i)})$      ▷ (16)
    retrieve $\mu_\theta(z^{(i)})$
    sample $x^{(i)}_{\text{recon}} \sim p_{\theta,\nu}(x|z^{(i)})$      ▷ (15)

**if** TRAINING PHASE **then**
    Take gradient step with $\nabla_{\phi,\theta}\mathcal{L}_\gamma(\phi,\theta)$
    using t-reparametrization trick      ▷ (24)
**if** GENERATION PHASE **then**
    sample $z_{\text{gen}} \sim p_\nu^\star(z)$      ▷ (19)
    retrieve $\mu_\theta(z_{\text{gen}})$
    sample $x_{\text{gen}} \sim p_{\theta,\nu}(x|z_{\text{gen}})$      ▷ (15)

---



Figure 2: Log-histograms of samples generated from $t^3$VAE ($\nu = 9, 12, 15, 18, 21$), Gaussian VAE, $\beta$-VAE ($\beta = 0.1$), Student-$t$ VAE, DE-VAE and VAE-st. Solid lines illustrate the true density $p_{\text{heavy}}$.

Bayesian model by introducing a *latent precision* $\lambda$ affecting both $x$ and $z$[2]:

$$\lambda \sim \chi^2(\lambda|\nu), \quad z|\lambda \sim \mathcal{N}_m(z|0, \nu\lambda^{-1}I), \quad x|z,\lambda \sim \mathcal{N}_n(x|\mu_\theta(z), \nu\lambda^{-1}\sigma^2 I). \quad (21)$$

It is then straightforward to add any number of latent layers $z_i|z_{<i}, \lambda \sim \mathcal{N}_{m_i}(z_i|\mu_\theta(z_{<i}), \nu\lambda^{-1}\sigma_i^2 I)$ to obtain a heavy-tailed hierarchical prior $(z_1, \cdots, z_L)$. In Appendix B.5, we develop a two-level hierarchical $t^3$VAE or $t^3$HVAE, where the priors $z_1$ and $z_2|z_1$, decoder $x|(z_1, z_2)$ and encoders $z_1|x$ and $z_2|(x, z_1)$ are all t-distributed. We also rederive the $\gamma$-loss from the information geometric formulation (17), showcasing the applicability of our approach.

## 4 EXPERIMENTS

In this section, we explore the empirical advantages that $t^3$VAE offers over the Gaussian VAE and other alternative models by analyzing various aspects of performance on synthetic and real datasets. A summary of our framework is provided in Algorithm 1 to assist with implementation. Experimental details including the reparametrization trick for $t^3$VAE, network architectures and further results are documented in Appendix C.

### 4.1 LEARNING HEAVY-TAILED BIMODAL DISTRIBUTIONS

**Univariate dataset.** We begin by analyzing generative performance on a univariate heavy-tailed dataset. We compare $t^3$VAE with $\nu \in \{9, 12, 15, 18, 21\}$, Gaussian VAE and $\beta$-VAE, as well as other t-based models: Student-$t$ VAE (Takahashi et al., 2018), DE-VAE (Mathieu et al., 2019) and VAE-st (Abiri & Ohlsson, 2020). Each model is trained on 200K samples from the bimodal density

$$p_{\text{heavy}}(x) = 0.6 \times t_1(x| - 2, 1^2, 5) + 0.4 \times t_1(x|2, 1^2, 5). \quad (22)$$

We then generate 500K samples from each of the trained models and compare to $p_{\text{heavy}}$, illustrated in Figure 2. We plot log-scale histograms in order to capture behavior in low-density regions. The Gaussian and $\beta$-VAE completely fail to produce reliable tail samples, in particular generating none beyond the range of $\pm 10$. In contrast, t-based models (with the exception of VAE-st) are capable of learning a much wider range of tail densities, establishing the efficacy of heavy-tailed models. For

---

[2]The factor $\frac{1+\nu^{-1}\|z\|^2}{1+\nu^{-1}m}$ in the t-decoder (15) may hence be understood as incorporating information gained on the latent precision when $z$ is observed. For tail region values encoded with large magnitude, one infers a smaller precision and thus increased variance for the output.

Table 1: $p$-values for the MMD test for (a) univariate and (b) bivariate synthetic heavy-tailed data. Rejected values are shown in red. Hyperparameters are tuned separately for each model and the best versions are reported; see Tables C1 and C2 for the full data.

| Model | Full | Left tail | Right tail | Model | Full | Left tail | Right tail |
|---|---|---|---|---|---|---|---|
| $t^3$VAE ($\nu = 18$) | 0.322 | 0.377 | 0.693 | Student $t$-VAE | 0.587 | 0.291 | 0.114 |
| VAE | 0.514 | 0.036 | 0.003 | DE-VAE ($\nu = 9$) | 0.943 | 0.424 | 0.814 |
| $\beta$-VAE ($\beta = 0.1$) | 0.614 | 0.011 | $< 0.001$ | VAE-st ($\nu = 12$) | $< 0.001$ | 0.953 | 0.643 |

(a) MMD test $p$-values for univariate data.

| Model | Full | Left tail | Right tail | Model | Full | Left tail | Right tail |
|---|---|---|---|---|---|---|---|
| $t^3$VAE ($\nu = 30$) | 0.276 | 0.214 | 0.213 | Student $t$-VAE | 0.530 | $< 0.001$ | $< 0.001$ |
| VAE | 0.116 | 0.004 | $< 0.001$ | DE-VAE ($\nu = 3$) | 0.624 | 0.002 | 0.057 |
| $\beta$-VAE ($\beta = 0.5$) | 0.251 | 0.011 | $< 0.001$ | VAE-st ($\nu = 3$) | 0.485 | $< 0.001$ | $< 0.001$ |

(b) MMD test $p$-values for bivariate data.

$t^3$VAE, we observe the best performance for $\nu \approx 20$; smaller $\nu$ leads to tail overestimation, while further increasing $\nu$ loses mass and ultimately converges to the Gaussian case as $\nu \to \infty$.

For a quantitative comparison, we apply the maximum mean discrepancy (MMD) test (Gretton et al., 2012) to evaluate whether the generated distributions are distinguishable from the original. As the test is insensitive to tail behavior, we also truncate the data by removing all points with absolute value less than 6 and apply the MMD test to the resulting tails. Table 1(a) shows $p$-values for the full dataset, left ($x < -6$) and right tails ($x > 6$). Testing with the full dataset fails to reject most models; restricted to low-density regions, however, MMD testing completely rejects the Gaussian VAE and tuned $\beta$-VAE. In contrast, our model output is indistinguishable from $p_{\text{heavy}}$ for moderate values of $\nu$.

**Bivariate dataset.** To distinguish $t^3$VAE from other t-based models, we further design a bivariate heavy-tailed dataset. We generate 200K samples $x_i$ from $p_{\text{heavy}}(x) = 0.7 \times t_1(x \mid -2, 2^2, 5) + 0.3 \times t_1(x \mid 2, 2^2, 5)$ and take $y_i = x_i + 2 \sin \left( \frac{\pi}{4} x_i \right)$ with noise distributed as $t_2((0,0)^\top, I_2, 6)$. We then run the MMD test for the full data and the left ($x^2 + y^2 > 10^2, x < 0$) and right ($x^2 + y^2 > 10^2, x > 0$) tail regions. The results are presented in Table 1(b). All models approximate the true distribution well in high-density regions; however, for low-density generation, the null hypothesis is rejected in every case except for $t^3$VAE, demonstrating the utility of implementing the *multivariate* t-distribution.

## 4.2 Learning High-dimensional Images

We now showcase the effectiveness of our model on high-dimensional data via both reconstruction and generation tasks in CelebA (Liu et al., 2015; 2018). The Fréchet inception distance (FID) score (Heusel et al., 2017) is employed to evaluate image quality. In order to comprehensively establish superiority, we also compare against a wide range of recent alternative VAE models.

A natural question to ask is whether heavy-tailedness is indeed the factor contributing to effective latent encoding. We address these concerns by comparing with simpler alternatives: a Gaussian prior with increased variance $\mathcal{N}_m(0, \kappa^2 I)$ for $\kappa > 1$, and $\beta$-VAE with regularizer weighting $\beta < 1$. We also include Student-$t$ VAE and DE-VAE as an ablation study of t-based model components. In addition, we implement two strong models which address latent structural collapse from different perspectives. The Tilted VAE (Floto et al., 2023), whose prior is designed to contain more mass in high-density rather than low-density regions; and FactorVAE (Kim & Mnih, 2018), to assess the effect of disentanglement. Hyperparameters of all models are tuned separately for each experiment.

**CelebA reconstruction.** We evaluate the quality of reconstructed $64 \times 64$ images in Table 2(a), where $t^3$VAE consistently achieves the lowest FID out of all models. While Tilted VAE and $\beta$-VAE also show good performance, they still cannot match $t^3$VAE, in particular for rarer class labels. These images exist in low-density regions of the data distribution; therefore, we hypothesize that the heavy-tailedness of $t^3$VAE makes it better suited in particular to learning rare features. We note that

Table 2: Reconstruction FID scores of CelebA and CIFAR100-LT. In CelebA, both overall scores and selected classes are shown. Bald, Mustache (Mst), and Gray hair (Gray) are rare classes (less than 5% of the total), while No beard (No Bd) is common (over 50%). In CIFAR100-LT, FID is measured varying imbalance factor $\rho$. Complete results of tuning each model are included in Appendix C.3.

(a) CelebA

| Framework | All | Bald | Mst | Gray | No Bd |
|---|---|---|---|---|---|
| $t^3$VAE ($\nu = 10$) | **39.4** | **66.5** | **61.5** | **67.2** | **40.1** |
| VAE | 57.9 | 85.8 | 79.7 | 91.0 | 58.4 |
| VAE ($\kappa = 1.5$) | 73.2 | 105.3 | 96.4 | 114.5 | 73.8 |
| $\beta$-VAE ($\beta = 0.05$) | 40.4 | 69.3 | 62.7 | 71.1 | 40.9 |
| Student-$t$ VAE | 78.4 | 112.0 | 104.2 | 118.7 | 78.6 |
| DE-VAE ($\nu = 5$) | 58.9 | 89.6 | 84.3 | 94.9 | 59.1 |
| Tilted VAE ($\tau = 50$) | 42.6 | 73.0 | 65.4 | 73.7 | 42.9 |
| FactorVAE ($\gamma_{\text{tc}} = 5$) | 59.8 | 91.7 | 85.7 | 95.2 | 60.8 |

(b) CIFAR100-LT

| Framework | $\rho = 1$ | 10 | 50 | 100 |
|---|---|---|---|---|
| $t^3$VAE ($\nu = 10$) | **97.5** | **102.8** | **108.3** | **128.7** |
| VAE | 256.1 | 267.2 | 277.4 | 287.3 |
| VAE ($\kappa = 1.5$) | 274.2 | 290.5 | 296.7 | 297.7 |
| $\beta$-VAE ($\beta = 0.1$) | 114.1 | 130.4 | 138.5 | 160.6 |
| Student-$t$ VAE | 259.5 | 314.1 | 323.7 | 333.4 |
| DE-VAE ($\nu = 2.5$) | 219.4 | 250.2 | 256.7 | 258.5 |
| Tilted VAE ($\tau = 50$) | 101.0 | 126.1 | 147.0 | 193.2 |
| FactorVAE ($\gamma_{\text{tc}} = 5$) | 232.3 | 272.5 | 275.6 | 270.1 |

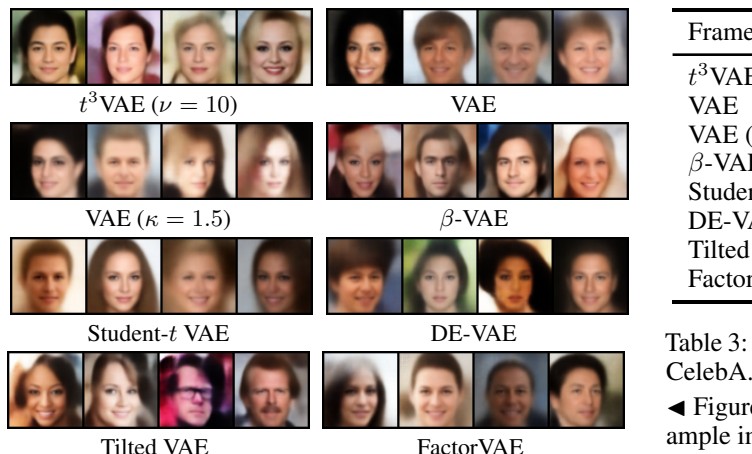

| Original | $t^3$VAE | VAE | VAE ($\kappa = 1.5$) | Tilted VAE |

Figure 3: Original and reconstructed images by $t^3$VAE ($\nu = 10$), Gaussian VAE, VAE with $\kappa = 1.5$, and Tilted VAE ($\tau = 50$).

simply increasing the variance of a Gaussian VAE results in images with reduced clarity. Moreover, disentangling does not seem to significantly alleviate over-regularization.

Investigating further, the images in Figure 3 display rare feature combinations; for example, the top left image belongs to the intersection of the Male and Heavy Make-up classes, which constitute around 1% of all images. We see that the Gaussian VAE largely fails to learn the image and instead generates a much more feminine face, evidenced by e.g. eyeshadow and lip color. In contrast, our model is able to more closely mirror the original image. Hence, $t^3$VAE is capable of learning images more accurately, in particular those with rare features.

| $t^3$VAE ($\nu = 10$) | VAE |
| VAE ($\kappa = 1.5$) | $\beta$-VAE |
| Student-$t$ VAE | DE-VAE |
| Tilted VAE | FactorVAE |

| Framework | FID |
|---|---|
| $t^3$VAE ($\nu = 10$) | **50.6** |
| VAE | 64.7 |
| VAE ($\kappa = 1.5$) | 79.6 |
| $\beta$-VAE ($\beta = 0.05$) | 51.8 |
| Student-$t$ VAE | 82.3 |
| DE-VAE ($\nu = 2.5$) | 58.9 |
| Tilted VAE ($\tau = 30$) | 59.2 |
| FactorVAE ($\gamma_{\text{tc}} = 2.5$) | 67.0 |

Table 3: Generation FID scores for CelebA.

◀ Figure 4: Generated CelebA example images.

**CelebA generation.** Works on VAE models often do not consider the generative aspect due to difficulties in producing sharp images. Nevertheless, we find that $t^3$VAE consistently generates high-quality samples if we sample from the *alternative* t-prior $p_\nu^\star(z)$. As demonstrated in Table 3 and Figure 4, $t^3$VAE outperforms all other models in FID score and generates more vivid images. We note that $\beta$-VAE is a close contender but cannot surpass $t^3$VAE even when $\beta$ is fine-tuned; generation FID scores for various $\beta$ values are documented in Appendix C.3.

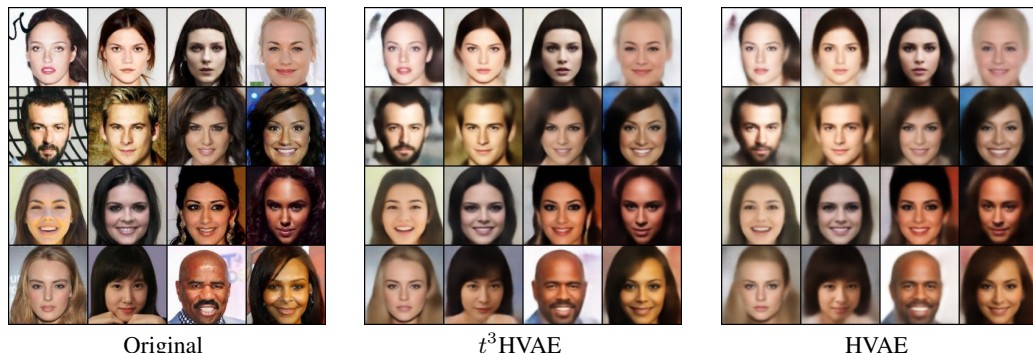

| Original | $t^3$HVAE | HVAE |

Figure 5: Original and reconstructed images by $t^3$HVAE ($\nu = 10$) and HVAE.

**Imbalanced CIFAR.** One interpretation of the heavy-tailedness of real-world data is when the occurrence frequencies of each class follow a long-tailed distribution. For instance, CelebA comprises facial attribute labels with highly varying frequencies. Motivated by this observation, we conduct reconstruction experiments with the CIFAR100-LT dataset (Cao et al., 2019), which is a long-tailed version of the original CIFAR-100 (Krizhevsky, 2009). We further consider varying degrees of *imbalance*, quantified by the imbalance factor $\rho$ which is defined as the ratio of instances in the largest class to the smallest. In this experiment, we take $\rho \in \{1, 10, 50, 100\}$ by linearly reducing the number of instances in each class.

Table 2(b) reports reconstruction FID scores of each tuned model, where $t^3$VAE again achieves the lowest scores. This time, we draw the reader's attention to the comparative performance of $t^3$VAE, $\beta$-VAE and Tilted VAE as $\rho$ varies. While the scores are relatively similar in the balanced case, the gaps become larger as $\rho$ increases; that is, $t^3$VAE experiences less of a performance drop even for extremely lopsided datasets ($\rho = 100$). In conclusion, we verify that $t^3$VAE is especially strong with regard to imbalanced data with long tails.

$t^3$**HVAE.** We additionally implement the two-layer hierarchical $t^3$HVAE constructed in Appendix B.5 and compare with the Gaussian HVAE on higher resolution $128 \times 128$ CelebA images. The reconstruction results are displayed in Figure 5, and corresponding FID scores are recorded in Table C3. We see that the increased hierarchical depth allows $t^3$HVAE to learn more sophisticated images, again with substantially higher clarity and sharper detail compared to the Gaussian HVAE. These results further justify the generality and effectiveness of our theoretical framework.

**Training cost and hyperparameter selection.** Unlike models such as DE-VAE or VAE-st which require numerical integration to calculate KL divergence between t-distributions for each data point, the explicit form of the $\gamma$-loss (18), as well as the t-based sampling and reparametrization processes, do not create any computational bottlenecks or instability issues; the runtime of $t^3$VAE was virtually identical to the corresponding Gaussian VAE. In addition, as shown in Figure 1 and also corroborated by experiments, $t^3$VAE's performance remains consistent across different values of $\nu$ for high-dimensional data. Thus another unique strength of our model is that there is no need to extensively tune the hyperparameter; only a couple of trials suffice.

## 5 CONCLUSION

Motivated by topics in over-regularization and information geometry, we establish a novel VAE framework for approximating heavy-tailed densities that uses Student's t-distributions for prior, encoder and decoder. We also derive the $\gamma$-loss objective by replacing the KL divergence joint optimization formulation of ELBO with $\gamma$-power divergence, and study the effects of its regularizer. The generative and tail-modeling capabilities of $t^3$VAE are demonstrated on heavy-tailed synthetic data. Finally, we show that $t^3$VAE outperforms various alternative VAE models and can learn images in richer detail, especially in the presense of rare classes or imbalance. Our ideas may hopefully be extended to other probabilistic or divergence-based inference models in future works.

## ACKNOWLEDGMENTS

This work was supported by the AI-Bio Research Grant through Seoul National University and by the Institute of Information & Communications Technology Planning & Evaluation (IITP) grant funded by the South Korean government (MSIT) [NO.2021-0-01343-004, Artificial Intelligence Graduate School Program (Seoul National University)].

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

# APPENDIX

In Appendices A and B we give precise statements, proofs and more details for the theory presented in Sections 2 and 3. The argument in A.1 is due to Han et al. (2020). The material in A.2 up to Proposition 2 is adapted from Eguchi (2021), while the general theory of information geometry is presented in Amari (2016). All subsequent material is our own original work. In Appendix C, we provide implementation and training details for the experiments conducted in our paper.

## A  THEORETICAL DETAILS

### A.1  REWRITING THE ELBO AS KL DIVERGENCE

Equation (5) states that minimizing the divergence between points on the model and data distribution manifolds is equivalent to maximizing the expected ELBO. This can be shown by performing the following algebraic manipulation:

$$
\begin{aligned}
\mathcal{D}_{\mathrm{KL}}(q_\phi(x,z) \,\|\, p_\theta(x,z)) &= \iint p_{\mathrm{data}}(x)q_\phi(z|x) \log \frac{p_{\mathrm{data}}(x)q_\phi(z|x)}{p_\theta(x)p_\theta(z|x)} dz dx \\
&= \int p_{\mathrm{data}}(x) \log \frac{p_{\mathrm{data}}(x)}{p_\theta(x)} dx + \int p_{\mathrm{data}}(x) \left[ \int q_\phi(z|x) \log \frac{q_\phi(z|x)}{p_\theta(z|x)} dz \right] dx \\
&= \mathbb{E}_{x \sim p_{\mathrm{data}}} \left[ -\log p_\theta(x) + \mathcal{D}_{\mathrm{KL}}(q_\phi(z|x) \,\|\, p_\theta(z|x)) \right] - H(p_{\mathrm{data}}) \\
&= -\mathbb{E}_{x \sim p_{\mathrm{data}}} \left[ \mathcal{L}(x; \theta, \phi) \right] - H(p_{\mathrm{data}}).
\end{aligned}
$$

This formulation opens the door to various VAE modifications by replacing the joint model KL divergence with other divergences, assuming closed-form computation is possible and results in an estimable objective.

### A.2  INFORMATION GEOMETRY OF $\gamma$-POWER DIVERGENCE

Let $\mathcal{M} = \{p_\theta(x) : \theta \in \Theta\}$ be a statistical manifold of probability distributions on a probability space $X$ parametrized by local coordinates $\theta = (\theta_1, \cdots, \theta_d)^\top$. A divergence on $\mathcal{M}$ is a $C^2$ function $\mathcal{D} : \mathcal{M} \times \mathcal{M} \to \mathbb{R}_{\geq 0}$ satisfying

1. $\mathcal{D}(q \,\|\, p) \geq 0$ for all $p, q \in \mathcal{M}$,
2. $\mathcal{D}(q \,\|\, p) = 0$ if and only if $p = q$,
3. At any point $p_\theta \in \mathcal{M}$, $\mathcal{D}(p_\theta \,\|\, p_{\theta'})$ is a positive-definite quadratic form for infinitesimal displacements $\theta' = \theta + d\theta$,

$$
\mathcal{D}(p_\theta \,\|\, p_{\theta'}) = \frac{1}{2} \sum_{i,j=1}^d g_{ij}(\theta) d\theta_i d\theta_j + O(\|d\theta\|^3)
$$

for a symmetric positive-definite matrix $g(\theta) = (g_{ij}(\theta))_{1 \leq i,j \leq d}$.

**Proposition 1.** *$\gamma$-power divergence as defined in Equations (7)-(8) for $\gamma \in (-1, 0) \cup (0, \infty)$ is a divergence on the finite $\gamma$-entropy submanifold $\{p \in \mathcal{M} : \|p\|_{1+\gamma} < \infty\}$ of $\mathcal{M}$.*

*Proof.* When $\gamma > 0$, we have by Hölder's inequality

$$
\int q(x)p(x)^\gamma dx \leq \|q\|_{1+\gamma} \|p^\gamma\|_{1+1/\gamma} = \|q\|_{1+\gamma} \|p\|_{1+\gamma}^\gamma,
$$

with equality iff $p = q$. When $-1 < \gamma < 0$, we have by the reverse Hölder inequality

$$
\int q(x)p(x)^\gamma dx \geq \|q\|_{1+\gamma} \|p^\gamma\|_{1+1/\gamma} = \|q\|_{1+\gamma} \|p\|_{1+\gamma}^\gamma,
$$

again with equality iff $p = q$. In both cases we conclude that

$$
\mathcal{D}_\gamma(q \,\|\, p) = -\frac{1}{\gamma} \int q(x) \left( \frac{p(x)}{\|p\|_{1+\gamma}} \right)^\gamma dx + \frac{1}{\gamma} \|q\|_{1+\gamma} \geq 0.
$$

The 1st and 2nd order local expansion terms of $\mathcal{D}_\gamma(p_\theta \,\|\, p_{\theta'})$ around $\theta' = \theta$ can be computed as

$$\frac{\partial}{\partial \theta_i}\bigg|_{\theta'=\theta} \mathcal{D}_\gamma(p_\theta \,\|\, p_{\theta'}) = -\frac{1}{\gamma} \int \frac{\partial p_\theta}{\partial \theta_i}(x) \left( \frac{p_\theta(x)}{\|p_\theta\|_{1+\gamma}} \right)^\gamma dx + \frac{1}{\gamma} \|p_\theta\|_{1+\gamma}^{-\gamma} \int p_\theta(x)^\gamma \frac{\partial p_\theta}{\partial \theta_i}(x) dx = 0,$$

$$\frac{\partial}{\partial \theta_i'}\bigg|_{\theta'=\theta} \mathcal{D}_\gamma(p_\theta \,\|\, p_{\theta'}) = -\int p_\theta(x) \left( \frac{p_\theta(x)}{\|p_\theta\|_{1+\gamma}} \right)^{\gamma-1} \frac{\partial}{\partial \theta_i} \left( \frac{p_\theta(x)}{\|p_\theta\|_{1+\gamma}} \right) dx = 0,$$

and

$$g_{ij}(\theta) = -\frac{\partial^2}{\partial \theta_i \partial \theta_j'}\bigg|_{\theta'=\theta} \mathcal{D}_\gamma(p_\theta \,\|\, p_{\theta'})$$

$$= \int \frac{\partial p_\theta}{\partial \theta_i}(x) \left( \frac{p_\theta(x)}{\|p_\theta\|_{1+\gamma}} \right)^{\gamma-1} \frac{\partial}{\partial \theta_j} \left( \frac{p_\theta(x)}{\|p_\theta\|_{1+\gamma}} \right) dx$$

$$= \|p_\theta\|_{1+\gamma}^{-1-2\gamma} \left[ \left( \int p_\theta(x)^{1+\gamma} dx \right) \left( \int p_\theta(x)^{\gamma-1} \frac{\partial p_\theta}{\partial \theta_i}(x) \frac{\partial p_\theta}{\partial \theta_j}(x) dx \right) \right.$$

$$\left. - \left( \int p_\theta(x)^\gamma \frac{\partial p_\theta}{\partial \theta_i}(x) dx \right) \left( \int p_\theta(x)^\gamma \frac{\partial p_\theta}{\partial \theta_j}(x) dx \right) \right]$$

from which we can check that $g_{ij}(\theta) = g_{ji}(\theta)$. $\qquad\square$

Any divergence $\mathcal{D}$ on $\mathcal{M}$ thus naturally induces a Riemannian metric $g$ on $\mathcal{M}$. For KL divergence, this is the Fisher information metric. $\mathcal{D}$ also induces two affine connections $\Gamma, \Gamma^*$ on $\mathcal{M}$ as

$$\Gamma_{ij}^k(\theta) = -\frac{\partial^3}{\partial \theta_i \partial \theta_j \partial \theta_k'}\bigg|_{\theta'=\theta} \mathcal{D}(p_\theta \,\|\, p_{\theta'}), \quad \Gamma_{ij}^{*k}(\theta) = -\frac{\partial^3}{\partial \theta_i' \partial \theta_j' \partial \theta_k}\bigg|_{\theta'=\theta} \mathcal{D}(p_\theta \,\|\, p_{\theta'}).$$

The connections are dually coupled with respect to $g$, i.e. the parallel transport of any two vectors, each by $\Gamma$ and $\Gamma^*$, preserves their inner product. For $\gamma$-power divergence, it is straightforward to check that $\Gamma_{ij}^k(\theta)$ vanish identically for mixture families

$$p_\theta(x) = \sum_{i=1}^d \theta_i p^{(i)}(x) + \left( 1 - \sum_{i=1}^d \theta_i \right) p^{(0)}(x).$$

Hence, the $\Gamma$-autoparallel curve connecting two points is equivalent to the linearly interpolating $m$-geodesic, identically to KL divergence. Moreover, the dual symbols

$$\Gamma_{ij}^{*k}(\theta) = \frac{1}{\gamma} \int \frac{\partial p_\theta}{\partial \theta_i}(x) \frac{\partial^2}{\partial \theta_j \partial \theta_k} \left( \frac{p_\theta(x)}{\|p_\theta\|_{1+\gamma}} \right)^\gamma dx$$

vanish when $p_\theta(x)^\gamma$ is linear in $\theta$. Hence the $\Gamma^*$-autoparallel curve from distribution $p$ to $q$ is given by the $\gamma$-power geodesic

$$p_t(x) \propto \left[ (1-t) \left( \frac{p(x)}{\|p\|_{1+\gamma}} \right)^\gamma + t \left( \frac{q(x)}{\|q\|_{1+\gamma}} \right)^\gamma \right]^{\frac{1}{\gamma}}, \quad t \in [0,1].$$

Moreover, the totally $\Gamma^*$-geodesic submanifolds consist of power families (9) and in particular t-distributions with $\gamma = -\frac{2}{\nu+d}$. The power form can also be recovered as the maximal $\gamma$-entropy distributions.

**Proposition 2.** *For any statistic $s_0(x)$, a maximal $\gamma$-entropy distribution $p$ satisfying the mean constraint $\mathbb{E}_{x\sim p}[s_0(x)] = \mu$ is of the form of Equation (9).*

*Proof.* Introducing a multiplier $\lambda$, we seek the stationary points of the Lagrangian functional

$$\mathcal{J}(p, \lambda) = \int p(x)^{1+\gamma} dx + \lambda^\top \int (s_0(x) - \mu) p(x) dx.$$

The corresponding Euler-Lagrange equation is

$$(1+\gamma) p(x)^\gamma + \lambda^\top (s_0(x) - \mu) = 0,$$

which yields Equation (9) after a suitable scaling with $s(x) \propto s_0(x)$. $\qquad\square$

We proceed to analyze the $\gamma$-power divergence between two t-distributions.

**Proposition 3.** *The $\gamma$-power divergence from $q_\nu \sim t_d(\mu_0, \Sigma_0, \nu)$ to $p_\nu \sim t_d(\mu_1, \Sigma_1, \nu)$ is given by Equation (11) when $\nu > 2$ and $\gamma = -\frac{2}{\nu+d}$.*

*Proof.* We first evaluate the power integral

$$\int q_\nu(x) p_\nu(x)^\gamma dx = \mathbb{E}_{x \sim q_\nu} \left[ p_\nu(x)^\gamma \right]$$

$$= \mathbb{E}_{x \sim q_\nu} \left[ C_{\nu,d}^\gamma |\Sigma_1|^{-\frac{\gamma}{2}} \left( 1 + \frac{1}{\nu}(x-\mu_1)^\top \Sigma_1^{-1}(x-\mu_1) \right) \right]$$

$$= C_{\nu,d}^\gamma |\Sigma_1|^{-\frac{\gamma}{2}} \left( 1 + \frac{1}{\nu} \mathbb{E}_{x \sim q_\nu} \left[ \text{tr} \left( \Sigma_1^{-1}(x-\mu_1)(x-\mu_1)^\top \right) \right] \right)$$

$$= C_{\nu,d}^\gamma |\Sigma_1|^{-\frac{\gamma}{2}} \left( 1 + \frac{1}{\nu} \text{tr} \left( \Sigma_1^{-1} \frac{\nu}{\nu-2}\Sigma_0 + \Sigma_1^{-1}(\mu_0-\mu_1)(\mu_0-\mu_1)^\top \right) \right)$$

$$= C_{\nu,d}^\gamma |\Sigma_1|^{-\frac{\gamma}{2}} \left( 1 + \frac{1}{\nu-2} \text{tr} \left( \Sigma_1^{-1}\Sigma_0 \right) + \frac{1}{\nu}(\mu_0-\mu_1)^\top \Sigma_1^{-1}(\mu_0-\mu_1) \right)$$

and similarly

$$\int q_\nu(x)^{1+\gamma} dx = C_{\nu,d}^\gamma |\Sigma_0|^{-\frac{\gamma}{2}} \left( 1 + \frac{d}{\nu-2} \right), \quad \int p_\nu(x)^{1+\gamma} dx = C_{\nu,d}^\gamma |\Sigma_1|^{-\frac{\gamma}{2}} \left( 1 + \frac{d}{\nu-2} \right).$$

The $\gamma$-power entropy and cross-entropy may then be computed as

$$\mathcal{H}_\gamma(q_\nu) = -\left( \int q_\nu(x)^{1+\gamma} dx \right)^{\frac{1}{1+\gamma}} = -C_{\nu,d}^{\frac{\gamma}{1+\gamma}} |\Sigma_0|^{-\frac{\gamma}{2(1+\gamma)}} \left( 1 + \frac{d}{\nu-2} \right)^{\frac{1}{1+\gamma}},$$

$$\mathcal{C}_\gamma(q_\nu, p_\nu) = -\left( \int q_\nu(x) p_\nu(x)^\gamma dx \right) \left( \int q(x)^{1+\gamma} dx \right)^{-\frac{\gamma}{1+\gamma}} = -C_{\nu,d}^{\frac{\gamma}{1+\gamma}} |\Sigma_1|^{-\frac{\gamma}{2(1+\gamma)}}$$

$$\times \left( 1 + \frac{d}{\nu-2} \right)^{-\frac{\gamma}{1+\gamma}} \left( 1 + \frac{1}{\nu-2} \text{tr} \left( \Sigma_1^{-1}\Sigma_0 \right) + \frac{1}{\nu}(\mu_0-\mu_1)^\top \Sigma_1^{-1}(\mu_0-\mu_1) \right),$$

which combine to give Equation (11). $\qquad \square$

**Proposition 4.** *$\gamma$-power divergence converges to KL divergence as $\gamma \to 0$ in the following cases.*

1. *For any two fixed distributions $p, q$, $\lim_{\gamma \to 0} \mathcal{D}_\gamma(q \,\|\, p) = \mathcal{D}_{\text{KL}}(q \,\|\, p)$.*

2. *For $q_\nu \sim t_d(\mu_0, \Sigma_0, \nu)$ and $p_\nu \sim t_d(\mu_1, \Sigma_1, \nu)$ with $\nu = -\frac{2}{\gamma} - d$ and limiting distributions $q_\infty \sim \mathcal{N}_d(\mu_0, \Sigma_0)$ and $p_\infty \sim \mathcal{N}_d(\mu_1, \Sigma_1)$, $\lim_{\gamma \to 0} \mathcal{D}_\gamma(q_\nu \,\|\, p_\nu) = \mathcal{D}_{\text{KL}}(q_\infty \,\|\, p_\infty)$.*

*Proof.* For 1, we may calculate the linearization of $\mathcal{C}_\gamma(q, p)$ in $\gamma$ via differential coefficients,

$$\lim_{\gamma \to 0} \frac{\mathcal{C}_\gamma(q, p) + 1}{\gamma} = -\int q(x) \lim_{\gamma \to 0} \frac{1}{\gamma} \left[ \left( \frac{p(x)}{\|p\|_{1+\gamma}} \right)^\gamma - 1 \right] dx$$

$$= -\int q(x) \left. \frac{\partial}{\partial \gamma} \right|_{\gamma=0} \left( \frac{p(x)}{\|p\|_{1+\gamma}} \right)^\gamma dx$$

$$= -\int q(x) \log p(x) dx = \mathcal{C}(q, p)$$

where $\mathcal{C}(q, p)$ is the ordinary cross-entropy. The same relation holds for entropy when $p = q$. Thus,

$$\lim_{\gamma \to 0} \mathcal{D}_\gamma(q \,\|\, p) = \lim_{\gamma \to 0} \frac{\mathcal{C}_\gamma(q, p) + 1}{\gamma} - \frac{\mathcal{H}_\gamma(q) + 1}{\gamma} = \mathcal{C}(q, p) - \mathcal{H}(q) = \mathcal{D}_{\text{KL}}(q \,\|\, p).$$

For 2, it is straightforward to show directly that Equation (11) converges to

$$\mathcal{D}_{\text{KL}}(q_\infty \,\|\, p_\infty) = \frac{1}{2} \left( \log \frac{|\Sigma_1|}{|\Sigma_0|} - d + \text{tr} \left( \Sigma_1^{-1}\Sigma_0 \right) + (\mu_0-\mu_1)^\top \Sigma_1^{-1}(\mu_0-\mu_1) \right)$$

by noting that $C_{\nu,d} \sim (2\pi^2\nu)^{-d/2}$ from Stirling's approximation and $\gamma\nu \to -2$. $\qquad \square$

## B  $t^3$VAE COMPUTATIONS

### B.1  DERIVATION OF JOINT MODEL AND BAYESIAN VIEW

The joint model distribution (13) of $t^3$VAE is explicitly defined as

$$p_{\theta,\nu}(x,z) = C_{\nu,m+n}\sigma^{-n}\left[1 + \frac{1}{\nu}\left(\|z\|^2 + \frac{1}{\sigma^2}\|x - \mu_\theta(z)\|^2\right)\right]^{-\frac{\nu+m+n}{2}}.$$

We may retrieve the prior (14) by marginalizing out $x$, which also confirms $p_{\theta,\nu}$ is a valid density:

$$p_{Z,\nu}(z) = \int p_{\theta,\nu}(x,z)dx$$

$$= C_{\nu,m+n}\sigma^{-n}\left(1 + \frac{1}{\nu}\|z\|^2\right)^{-\frac{\nu+m+n}{2}}\int\left(1 + \frac{1+\nu^{-1}m}{1+\nu^{-1}\|z\|^2}\frac{\|x-\mu_\theta(z)\|^2}{(\nu+m)\sigma^2}\right)^{-\frac{\nu+m+n}{2}}dx$$

$$= C_{\nu,m+n}\sigma^{-n}\left(1 + \frac{1}{\nu}\|z\|^2\right)^{-\frac{\nu+m+n}{2}}C_{\nu+m,n}^{-1}\left(\frac{1+\nu^{-1}\|z\|^2}{1+\nu^{-1}m}\sigma^2\right)^{\frac{n}{2}}$$

$$= C_{\nu,m+n}C_{\nu+m,n}^{-1}\left(1 + \frac{m}{\nu}\right)^{-\frac{n}{2}}\left(1 + \frac{1}{\nu}\|z\|^2\right)^{-\frac{\nu+m}{2}}$$

$$= C_{\nu,m}\left(1 + \frac{1}{\nu}\|z\|^2\right)^{-\frac{\nu+m}{2}}$$

$$= t_m(z|0, I, \nu),$$

where we have used the fact that

$$C_{\nu,m+n} = \frac{\Gamma\left(\frac{\nu+m+n}{2}\right)}{\Gamma\left(\frac{\nu+m}{2}\right)\left((\nu+m)\pi\right)^{\frac{n}{2}}}\frac{\Gamma\left(\frac{\nu+m}{2}\right)}{\Gamma\left(\frac{\nu}{2}\right)(\nu\pi)^{\frac{m}{2}}}\left(1 + \frac{m}{\nu}\right)^{\frac{n}{2}} = C_{\nu+m,n}C_{\nu,m}\left(1 + \frac{m}{\nu}\right)^{\frac{n}{2}}.$$

Consequently, the $t^3$VAE decoder is derived as

$$p_{\theta,\nu}(x|z) = \frac{p_{\theta,\nu}(x,z)}{p_{Z,\nu}(z)}$$

$$= C_{\nu+m,n}\sigma^{-n}\left(\frac{1+\nu^{-1}\|z\|^2}{1+\nu^{-1}m}\right)^{-\frac{n}{2}}\left(1 + \frac{1+\nu^{-1}m}{1+\nu^{-1}\|z\|^2}\frac{\|x-\mu_\theta(z)\|^2}{(\nu+m)\sigma^2}\right)^{-\frac{\nu+m+n}{2}}$$

$$= t_n\left(x\left|\mu_\theta(z), \frac{1+\nu^{-1}\|z\|^2}{1+\nu^{-1}m}\sigma^2 I, \nu+m\right.\right).$$

We now prove that the prior-decoder pair is equivalent to the Bayesian hierarchical model (21). By integrating out the latent precision:

$$z \sim \int_0^\infty \mathcal{N}_m\left(z\left|0, \frac{1}{\nu^{-1}\lambda}I\right.\right)\chi^2(\lambda|\nu)d\lambda$$

$$\propto \int_0^\infty \exp\left(-\frac{\lambda}{2\nu}\|z\|^2 - \frac{\lambda}{2}\right)\lambda^{\frac{\nu}{2}+\frac{m}{2}-1}d\lambda$$

$$\propto \left(1 + \frac{1}{\nu}\|z\|^2\right)^{-\frac{\nu+m}{2}},$$

and

$$x|z \sim \int_0^\infty \mathcal{N}_n\left(x\left|\mu_\theta(z), \frac{1}{\nu^{-1}\lambda}\sigma^2 I\right.\right)\mathcal{N}_m\left(z\left|0, \frac{1}{\nu^{-1}\lambda}I\right.\right)\chi^2(\lambda|\nu)d\lambda$$

$$\propto \int_0^\infty \exp\left(-\frac{\lambda}{2\nu\sigma^2}\|x-\mu_\theta(z)\|^2 - \frac{\lambda}{2\nu}\|z\|^2 - \frac{\lambda}{2}\right)\lambda^{\frac{\nu}{2}+\frac{m}{2}+\frac{n}{2}-1}d\lambda$$

$$\propto \left(1 + \frac{1}{\nu}\|z\|^2 + \frac{1}{\nu\sigma^2}\|x - \mu_\theta(z)\|^2\right)^{-\frac{\nu+m+n}{2}}$$

$$\propto t_n\left(x\,\Big|\,\mu_\theta(z), \frac{1+\nu^{-1}\|z\|^2}{1+\nu^{-1}m}\sigma^2 I, \nu + m\right),$$

we recover the $t^3$VAE architecture.

## B.2 SHALLOW $t^3$VAE

We consider the simplest case when the decoder mean $\mu_\theta(z) = Wz + b$ is linear with parameters $\theta = (W, b)$, $W \in \mathbb{R}^{n \times m}$, $b \in \mathbb{R}^n$, which we call the 'shallow' $t^3$VAE. For completeness, we first describe the corresponding shallow Gaussian VAE as well. In this case, the joint model (12) with $\mu_\theta(z) = Wz + b$ is easily checked to be normally distributed as

$$\begin{pmatrix} x \\ z \end{pmatrix} \sim \mathcal{N}_{m+n}\left(\begin{pmatrix} b \\ 0 \end{pmatrix}, \begin{pmatrix} WW^\top + \sigma^2 I & W \\ W^\top & I \end{pmatrix}\right)$$

and the true posterior can be obtained exactly as

$$z|x \sim \mathcal{N}_m\left(W^\top(WW^\top + \sigma^2 I)^{-1}(x - b), I - W^\top(WW^\top + \sigma^2 I)^{-1}W\right).$$

On the other hand, the joint model (13) for the shallow $t^3$VAE is t-distributed:

$$\begin{pmatrix} x \\ z \end{pmatrix} \propto \left[1 + \frac{1}{\nu\sigma^2}\begin{pmatrix} x - b \\ z \end{pmatrix}^\top \begin{pmatrix} I & -W \\ -W^\top & W^\top W + \sigma^2 I \end{pmatrix}\begin{pmatrix} x - b \\ z \end{pmatrix}\right]^{-\frac{\nu+m+n}{2}}$$

$$\propto t_{m+n}\left(\begin{pmatrix} b \\ 0 \end{pmatrix}, \begin{pmatrix} WW^\top + \sigma^2 I & W \\ W^\top & I \end{pmatrix}, \nu\right).$$

Then the true posterior is in fact also t-distributed, but with degrees of freedom $\nu + n$ rather than $\nu$ (Ding, 2016): $z|x \sim t_m(\tilde{\mu}(x), \tilde{\Sigma}(x), \nu + n)$ with mean

$$\tilde{\mu}(x) = W^\top(WW^\top + \sigma^2 I)^{-1}(x - b)$$

and scale matrix

$$\tilde{\Sigma}(x) = \frac{1 + \nu^{-1}(x-b)^\top(WW^\top + \sigma^2 I)^{-1}(x-b)}{1 + \nu^{-1}n}(I - W^\top(WW^\top + \sigma^2 I)^{-1}W).$$

This motivates the definition of the $t^3$VAE encoder (16).

## B.3 DERIVATION OF $\gamma$-LOSS

Our goal is to derive the $\gamma$-power divergence between a point $p_{\theta,\nu}(x, z)$ on the model distribution manifold $\mathcal{P}_\nu$, given by Equation (13), and a point on the data distribution manifold $\mathcal{Q}_\nu$, given by

$$q_{\phi,\nu}(x, z) = p_{\text{data}}(x) \times t_m\left(z\,\Big|\,\mu_\phi(x), \frac{1}{1+\nu^{-1}n}\Sigma_\phi(x), \nu + n\right).$$

We begin by computing the required double integrals. First,

$$\iint p_{\theta,\nu}(x, z)^{1+\gamma}dxdz = \mathbb{E}_{z \sim p_{Z,\nu}}\mathbb{E}_{x \sim p_{\theta,\nu}(\cdot|z)}\left[p_{\theta,\nu}(x, z)^\gamma\right]$$

$$= \sigma^{-\gamma n}C_{\nu,m+n}^\gamma\mathbb{E}_{z \sim p_{Z,\nu}}\mathbb{E}_{x \sim p_{\theta,\nu}(\cdot|z)}\left[1 + \frac{1}{\nu}\left(\|z\|^2 + \frac{1}{\sigma^2}\|x - \mu_\theta(z)\|^2\right)\right]$$

$$= \sigma^{-\gamma n}C_{\nu,m+n}^\gamma\mathbb{E}_{z \sim p_{Z,\nu}}\left[1 + \frac{1}{\nu}\|z\|^2 + \frac{1}{\nu\sigma^2}\text{tr}\left(\frac{\nu+m}{\nu+m-2}\cdot\frac{1+\nu^{-1}\|z\|^2}{1+\nu^{-1}m}\sigma^2 I\right)\right]$$

$$= \sigma^{-\gamma n}C_{\nu,m+n}^\gamma\mathbb{E}_{z \sim p_{Z,\nu}}\left[1 + \frac{1}{\nu}\|z\|^2 + \frac{n}{\nu+m-2}\left(1 + \frac{1}{\nu}\|z\|^2\right)\right]$$

$$= \sigma^{-\gamma n} C_{\nu,m+n}^{\gamma} \left( 1 + \frac{m+n}{\nu-2} \right),$$

where we have repeatedly used the fact that

$$\mathbb{E}_{x \sim t_d(\mu,\Sigma,\nu)} \|x - \mu\|^2 = \mathrm{tr}\, \mathbb{E}_{x \sim t_d(\mu,\Sigma,\nu)}[(x-\mu)(x-\mu)^\top] = \frac{\nu}{\nu-2} \mathrm{tr}\, \Sigma.$$

Next, using the entropy computations in the proof of Proposition 3,

$$\iint q_{\phi,\nu}(x,z)^{1+\gamma} dx dz = \int \left( \int q_{\phi,\nu}(z|x)^{1+\gamma} dz \right) p_{\mathrm{data}}(x)^{1+\gamma} dx$$

$$= C_{\nu+n,m}^{\gamma} \left( 1 + \frac{n}{\nu} \right)^{\frac{\gamma m}{2}} \left( 1 + \frac{m}{\nu+n-2} \right) \int |\Sigma_\phi(x)|^{-\frac{\gamma}{2}} p_{\mathrm{data}}(x)^{1+\gamma} dx.$$

Moreover, we have

$$\iint q_{\phi,\nu}(x,z) p_{\theta,\nu}(x,z)^\gamma dx dz = \mathbb{E}_{x \sim p_{\mathrm{data}}} \mathbb{E}_{z \sim q_{\phi,\nu}(\cdot|x)} \left[ p_{\theta,\nu}(x,z)^\gamma \right]$$

$$= \sigma^{-\gamma n} C_{\nu,m+n}^{\gamma} \mathbb{E}_{x \sim p_{\mathrm{data}}} \mathbb{E}_{z \sim q_{\phi,\nu}(\cdot|x)} \left[ 1 + \frac{1}{\nu} \left( \|z\|^2 + \frac{1}{\sigma^2} \|x - \mu_\theta(z)\|^2 \right) \right]$$

$$= \sigma^{-\gamma n} C_{\nu,m+n}^{\gamma} \mathbb{E}_{x \sim p_{\mathrm{data}}} \left[ 1 + \frac{1}{\nu} \left( \|\mu_\phi(x)\|^2 \right. \right.$$

$$\left. \left. + \frac{\nu}{\nu+n-2} \mathrm{tr}\, \Sigma_\phi(x) + \frac{1}{\sigma^2} \mathbb{E}_{z \sim q_{\phi,\nu}(\cdot|x)} \|x - \mu_\theta(z)\|^2 \right) \right],$$

by utilizing the sum-of-squares decomposition $\|z\|^2 = \|\mu_\phi(x) + (z - \mu_\phi(x))\|^2$. The $\gamma$-power entropy of the joint data distribution is then obtained as

$$\mathcal{H}_\gamma(q_{\phi,\nu}) = - \left( \iint q_{\phi,\nu}(x,z)^{1+\gamma} dx dz \right)^{\frac{1}{1+\gamma}}$$

$$= - \underbrace{C_{\nu+n,m}^{\frac{\gamma}{1+\gamma}} \left( 1 + \frac{n}{\nu} \right)^{\frac{\gamma m}{2(1+\gamma)}} \left( 1 + \frac{m}{\nu+n-2} \right)^{\frac{1}{1+\gamma}}}_{=: C_1} \left( \int |\Sigma_\phi(x)|^{-\frac{\gamma}{2}} p_{\mathrm{data}}(x)^{1+\gamma} dx \right)^{\frac{1}{1+\gamma}},$$

and the $\gamma$-power cross-entropy is

$$\mathcal{C}_\gamma(q_{\phi,\nu}, p_{\theta,\nu}) = - \left( \iint q_{\phi,\nu}(x,z) p_{\theta,\nu}(x,z)^\gamma dx dz \right) \left( \iint p_{\theta,\nu}(x,z)^{1+\gamma} dx dz \right)^{-\frac{\gamma}{1+\gamma}}$$

$$= - \underbrace{\left( \sigma^{-\gamma n} C_{\nu,m+n}^{\gamma} \right)^{\frac{1}{1+\gamma}} \left( 1 + \frac{m+n}{\nu-2} \right)^{-\frac{\gamma}{1+\gamma}}}_{=: C_2}$$

$$\times \mathbb{E}_{x \sim p_{\mathrm{data}}} \left[ 1 + \frac{1}{\nu} \left( \frac{1}{\sigma^2} \mathbb{E}_{z \sim q_{\phi,\nu}(\cdot|x)} \|x - \mu_\theta(z)\|^2 + \|\mu_\phi(x)\|^2 + \frac{\nu}{\nu+n-2} \mathrm{tr}\, \Sigma_\phi(x) \right) \right].$$

Substituting the above two expressions in the definition of $\gamma$-power divergence (8) yields the formula

$$\mathcal{D}_\gamma(q_{\phi,\nu} \,\|\, p_{\theta,\nu}) = \frac{C_1}{\gamma} \left( \int |\Sigma_\phi(x)|^{-\frac{\gamma}{2}} p_{\mathrm{data}}(x)^{1+\gamma} dx \right)^{\frac{1}{1+\gamma}} - \frac{C_2}{\gamma}$$

$$\times \mathbb{E}_{x \sim p_{\mathrm{data}}} \left[ 1 + \frac{1}{\nu} \left( \frac{1}{\sigma^2} \mathbb{E}_{z \sim q_{\phi,\nu}(\cdot|x)} \|x - \mu_\theta(z)\|^2 + \|\mu_\phi(x)\|^2 + \frac{\nu}{\nu+n-2} \mathrm{tr}\, \Sigma_\phi(x) \right) \right].$$

We now provide justification for the approximation

$$\left( \int |\Sigma_\phi(x)|^{-\frac{\gamma}{2}} p_{\mathrm{data}}(x)^{1+\gamma} dx \right)^{\frac{1}{1+\gamma}} \simeq \int |\Sigma_\phi(x)|^{-\frac{\gamma}{2(1+\gamma)}} p_{\mathrm{data}}(x) dx - H(p_{\mathrm{data}}) \cdot \gamma$$

which is valid up to first order when $|\gamma| \ll 1$. Note that the linear coefficient $\mathcal{H}(p_{\mathrm{data}})$ is independent of parameters $\theta$ and $\phi$, leading to the $\gamma$-loss (18) after rearranging constants:

$$\mathcal{L}_\gamma(\theta, \phi) \approx - \frac{\gamma \nu}{2 C_2} \mathcal{D}_\gamma(q_{\phi,\nu} \,\|\, p_{\theta,\nu}) - \frac{\gamma \nu C_1}{2 C_2} \mathcal{H}(p_{\mathrm{data}}) - \frac{\nu}{2}.$$

**Proposition 5.** *Let $\sigma$ be any positive continuous function, $\gamma \in (-1, 0)$, and suppose the values*

$$H_{j,k} := \mathcal{H}_{j,k}(p_{\text{data}}) := \int p_{\text{data}}(x)^{1+j\gamma} |\log p_{\text{data}}(x)|^k \, dx$$

*are finite for each $j \in \{0, 1\}$, $k \in \{1, 2\}$. Then for any compact set $\Omega \subseteq \text{supp}\, p_{\text{data}}$,*

$$\left( \int_\Omega \sigma(x)^{-\gamma} p_{\text{data}}(x)^{1+\gamma} dx \right)^{\frac{1}{1+\gamma}} - \int_\Omega \sigma(x)^{-\frac{\gamma}{1+\gamma}} p_{\text{data}}(x) dx$$

$$= \gamma \int_\Omega p_{\text{data}}(x) \log p_{\text{data}}(x) dx + O(\gamma^2).$$

*Remark.* The assumption on $\sigma$ is valid since variance is typically trained by setting $\log \Sigma_\phi(x)$ as a neural network. The integrability condition on $p_{\text{data}}$ is slightly stronger than finite entropy and holds for e.g. $d$-dimensional Gaussian and t-distributions (when $\nu > 2$ and $\gamma = -\frac{2}{\nu + d}$).

*Proof.* Denote by $\sigma_{\min}, \sigma_{\max}$ the minimum and maximum of $\sigma$ on $\Omega$, respectively. We begin by linearizing the first term. Set $h(x) := \sigma(x)^{-1} p_{\text{data}}(x)$. For each $x \in \Omega$, we may Taylor expand

$$h(x)^\gamma = 1 + \gamma \log h(x) + \frac{1}{2} \gamma^2 \cdot (\log h(x))^2 h(x)^{\gamma^*(x)}$$

for some $\gamma^*(x) \in (\gamma, 0)$, and the integral of the remainder is bounded since

$$\int_\Omega (\log h(x))^2 h(x)^{\gamma^*(x)} p_{\text{data}}(x) dx$$

$$\leq \int_{\Omega \cap \{h \geq 1\}} (\log h(x))^2 p_{\text{data}}(x) dx + \int_{\Omega \cap \{h < 1\}} (\log h(x))^2 h(x)^\gamma p_{\text{data}}(x) dx$$

$$\leq \int_{\Omega \cap \{h \geq 1\}} (\log p_{\text{data}}(x) - \log \sigma_{\min})^2 p_{\text{data}}(x) dx$$

$$\quad + \sigma_{\max}^{-\gamma} \int_{\Omega \cap \{h < 1\}} (\log p_{\text{data}}(x) - \log \sigma_{\max})^2 p_{\text{data}}(x)^{1+\gamma} dx$$

$$\leq (\log \sigma_{\min})^2 + 2 |\log \sigma_{\min}| H_{0,1} + H_{0,2} + \sigma_{\max}^{-\gamma} \left( (\log \sigma_{\max})^2 + 2 |\log \sigma_{\max}| H_{1,1} + H_{1,2} \right)$$

$$< \infty.$$

Therefore, we obtain the approximation

$$\left( \int_\Omega h(x)^\gamma p_{\text{data}}(x) dx \right)^{\frac{1}{1+\gamma}} = 1 + \gamma \int_\Omega \log h(x) \cdot p_{\text{data}}(x) dx + O(\gamma^2).$$

For the second term, setting $h(x) = \sigma(x)^{-1}$ and linearizing with respect to $\eta := \frac{\gamma}{1+\gamma}$, we have

$$\int_\Omega \sigma(x)^{-\eta} p_{\text{data}}(x) dx = 1 - \eta \int_\Omega \log \sigma(x) \cdot p_{\text{data}}(x) dx + O(\eta^2)$$

with the remainder bounded by $\eta^2/2 \cdot \max\{(\log \sigma_{\max})^2, (\log \sigma_{\min})^2\} \max\{1, \sigma_{\min}^{-1}\}$. Plugging in $\eta = \gamma + O(\gamma^2)$ and subtracting both sides gives the desired approximation. $\square$

### B.4 DERIVATION OF $\gamma$-REGULARIZER

We compute the $\gamma$-power divergence from the encoder distribution (16) to the alternative prior

$$p_\nu^\star(z) = t_m(z | 0, \tau^2 I, \nu + n),$$

where the constant $\tau$ is yet to be determined. By Equation (11), we have for $\gamma = -\frac{2}{\nu + m + n}$,

$$\mathcal{D}_\gamma(q_{\phi, \nu} \| p_\nu^\star) =$$

$$-\frac{1}{\gamma} C_{\nu+n,m}^{\frac{\gamma}{1+\gamma}} \left( 1 + \frac{m}{\nu + n - 2} \right)^{-\frac{\gamma}{1+\gamma}} \left[ -\left( 1 + \frac{n}{\nu} \right)^{\frac{\gamma m}{2(1+\gamma)}} |\Sigma_\phi(x)|^{-\frac{\gamma}{2(1+\gamma)}} \left( 1 + \frac{m}{\nu + n - 2} \right) \right.$$

$$\left. + \frac{1}{\nu + n} \tau^{-2 - \frac{\gamma m}{1+\gamma}} \left( \|\mu_\phi(x)\|^2 + \frac{\nu}{\nu + n - 2} \operatorname{tr} \Sigma_\phi(x) + (\nu + n)\tau^2 \right) \right].$$

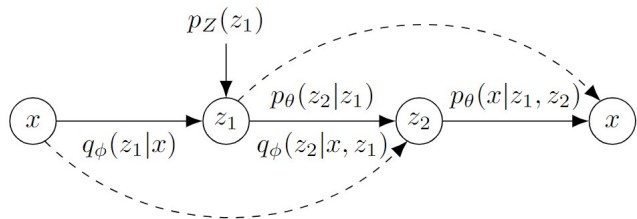

Figure B1: Structure of hierarchical $t^3$VAE.

Note that the coefficient of $\operatorname{tr}\Sigma_\phi(x)$ relative to the $\|\mu_\phi(x)\|^2$ term is $\frac{\nu}{\nu+n-2}$, aligning with the $\gamma$-loss formula (18). It remains to choose $\tau$ to match the coefficient $-\frac{\nu C_1}{C_2}$ of the $|\Sigma_\phi(x)|^{-\frac{\gamma}{2(1+\gamma)}}$ term:

$$-(\nu+n)\tau^{2+\frac{\gamma m}{1+\gamma}}\left(1+\frac{n}{\nu}\right)^{\frac{\gamma m}{2(1+\gamma)}}\left(1+\frac{m}{\nu+n-2}\right) = -\frac{\nu C_1}{C_2}$$

$$= -\nu\left(\frac{\nu+m+n-2}{\nu+n-2}\left(1+\frac{n}{\nu}\right)^{\frac{\gamma m}{2}}C_{\nu+n,m}^\gamma\right)^{\frac{1}{1+\gamma}}\left(\frac{\nu+m+n-2}{\nu-2}\sigma^n C_{\nu,m+n}^{-1}\right)^{\frac{\gamma}{1+\gamma}},$$

from which we obtain

$$\tau^2 = \frac{1}{1+\nu^{-1}n}\left(\sigma^{-n}C_{\nu,n}\left(1+\frac{n}{\nu-2}\right)^{-1}\right)^{\frac{2}{\nu+n-2}}.$$

Furthermore, the precise multiplicative factor required to match the $\gamma$-loss is (again looking at the $\|\mu_\phi(x)\|^2$ term)

$$\frac{1}{2}\times\left[-\frac{1}{\gamma}C_{\nu+n,m}^{\frac{\gamma}{1+\gamma}}\left(1+\frac{m}{\nu+n-2}\right)^{-\frac{\gamma}{1+\gamma}}\frac{1}{\nu+n}\tau^{-2-\frac{\gamma m}{1+\gamma}}\right]^{-1}$$

$$= -\frac{\gamma\nu}{2}C_{\nu+n,m}^{-\frac{\gamma}{1+\gamma}}\left(1+\frac{m}{\nu+n-2}\right)^{\frac{\gamma}{1+\gamma}}\left(1+\frac{n}{\nu}\right)^{-\frac{\gamma m}{2(1+\gamma)}}\sigma^{\frac{\gamma n}{1+\gamma}}C_{\nu,n}^{-\frac{\gamma}{1+\gamma}}\left(1+\frac{n}{\nu-2}\right)^{\frac{\gamma}{1+\gamma}}$$

$$= -\frac{\gamma\nu}{2}C_{\nu,m+n}^{-\frac{\gamma}{1+\gamma}}\left(1+\frac{m+n}{\nu-2}\right)^{\frac{\gamma}{1+\gamma}}\sigma^{\frac{\gamma n}{1+\gamma}}$$

$$= -\frac{\gamma\nu}{2C_2},$$

which is positive, implying that $\mathcal{D}_\gamma(q_{\phi,\nu}\,\|\,p_\nu^\star)$ indeed acts as a regularizer. Finally, the additive constant in Equation (20) is equal to $-\frac{1}{2}(\nu+n)\tau^2$.

## B.5 Constructing the $t^3$HVAE

We develop the heavy-tailed version of the two-level hierarchical VAE (Sønderby et al., 2016) with latent variables $(z_1, z_2) \in \mathbb{R}^{m_1+m_2}$, depicted in Figure B1. In the generative phase, $z_1$ is sampled from the initial prior $p_{Z,\nu}$, then $z_2$ is drawn from the conditional prior $p_{\theta,\nu}(z_2|z_1)$ and $x$ is drawn from the decoder $p_{\theta,\nu}(x|z_1, z_2)$. In the encoding phase, $z_1$ and $z_2$ are successively drawn from the hierarchical encoders $q_{\phi,\nu}(z_1|x)$ and $q_{\phi,\nu}(z_2|x, z_1)$, respectively. We remark that the second level decoder and encoder can also depend through skip connections on $z_1$ and $x$. Moreover, we assume two levels only for simplicity, and $t^3$HVAE can be readily expanded to any hierarchical depth.

The Bayesian scheme (21) naturally extends to

$$\lambda \sim \chi^2(\lambda|\nu), \quad z_1|\lambda \sim \mathcal{N}_{m_1}(z_1|0, \nu\lambda^{-1}I),$$

$$z_2|z_1, \lambda \sim \mathcal{N}_{m_2}(z_2|\zeta_\theta(z_1), \nu\lambda^{-1}\sigma_z^2 I), \quad x|z_1, z_2, \lambda \sim \mathcal{N}_n(z|\mu_\theta(z_1, z_2), \nu\lambda^{-1}\sigma_x^2 I),$$

which likewise to the derivation in Appendix B.1 integrates to give the t-priors and t-decoder:

$$p_{Z,\nu}(z_1) = t_{m_1}(z|0, I, \nu),$$

$$p_{\theta,\nu}(z_2|z_1) = t_{m_2}\left(z_2 \middle| \zeta_\theta(z_1), \frac{1+\nu^{-1}\|z_1\|^2}{1+\nu^{-1}m_1}\sigma_z^2 I, \nu+m_1\right),$$

$$p_{\theta,\nu}(x|z_1,z_2) = t_n\left(x \middle| \mu_\theta(z_1,z_2), \frac{1+\nu^{-1}\|z_1\|^2 + \nu^{-1}\sigma_z^{-2}\|z_2 - \zeta_\theta(z_1)\|^2}{1+\nu^{-1}(m_1+m_2)}\sigma_x^2 I, \nu+m_1+m_2\right).$$

We also define the t-encoder distributions as

$$q_{\phi,\nu}(z_1|x) = t_{m_1}\left(z_1 \middle| \zeta_\phi(x), \frac{1}{1+\nu^{-1}n}\Lambda_\phi(x), \nu+n\right),$$

$$q_{\phi,\nu}(z_2|x,z_1) = t_{m_2}\left(z_2 \middle| \mu_\phi(x,z_1), \frac{1}{1+\nu^{-1}(m_1+n)}\Sigma_\phi(x,z_1), \nu+m_1+n\right).$$

Defining $\gamma = -\frac{2}{\nu+m_1+m_2+n}$, the joint model distribution is derived as

$$p_{\theta,\nu}(x,z_1,z_2) = p_{Z,\nu}(z_1)p_{\theta,\nu}(z_2|z_1)p_{\theta,\nu}(x|z_1,z_2)$$

$$= C_{\nu,m_1}\left(1+\frac{1}{\nu}\|z_1\|^2\right)^{-\frac{\nu+m_1}{2}}$$

$$\times C_{\nu+m_1,m_2}\left(\frac{1+\nu^{-1}\|z_1\|^2}{1+\nu^{-1}m_1}\sigma_z^2\right)^{-\frac{m_2}{2}}\left(1+\frac{1}{\nu\sigma_z^2}\frac{\|z_2-\zeta_\theta(z_1)\|^2}{1+\nu^{-1}\|z_1\|^2}\right)^{-\frac{\nu+m_1+m_2}{2}}$$

$$\times C_{\nu+m_1+m_2,n}\left(\frac{1+\nu^{-1}\|z_1\|^2 + \nu^{-1}\sigma_z^{-2}\|z_2-\zeta_\theta(z_1)\|^2}{1+\nu^{-1}(m_1+m_2)}\sigma_x^2\right)^{-\frac{n}{2}}$$

$$\times\left(1+\frac{1}{\nu\sigma_x^2}\frac{\|x-\mu_\theta(z_2)\|^2}{1+\nu^{-1}\|z_1\|^2 + \nu^{-1}\sigma_z^{-2}\|z_2-\zeta_\theta(z_1)\|^2}\right)^{-\frac{\nu+m_1+m_2+n}{2}}$$

$$= C_{\nu,m_1}C_{\nu+m_1,m_2}C_{\nu+m_1+m_2,n}\sigma_z^{-m_2}\sigma_x^{-n}\left(1+\frac{m_1}{\nu}\right)^{\frac{m_2}{2}}\left(1+\frac{m_1+m_2}{\nu}\right)^{\frac{n}{2}}$$

$$\times\left(1+\frac{\|z_1\|^2}{\nu} + \frac{\|z_2-\zeta_\theta(z_1)\|^2}{\nu\sigma_z^2} + \frac{\|x-\mu_\theta(z_1,z_2)\|^2}{\nu\sigma_x^2}\right)^{-\frac{\nu+m_1+m_2+n}{2}}$$

$$= C_{\nu,m_1+m_2+n}\sigma_z^{-m_2}\sigma_x^{-n}\left(1+\frac{\|z_1\|^2}{\nu} + \frac{\|z_2-\zeta_\theta(z_1)\|^2}{\nu\sigma_z^2} + \frac{\|x-\mu_\theta(z_1,z_2)\|^2}{\nu\sigma_x^2}\right)^{\frac{1}{\gamma}}.$$

We proceed to derive the $\gamma$-power divergence between $p_{\theta,\nu}(x,z_1,z_2)$ and the data distribution $q_{\phi,\nu}(x,z_1,z_2) = q_{\phi,\nu}(z_2|x,z_1)q_{\phi,\nu}(z_1|x)p_{\text{data}}(x)$ analogously to Appendix B.3.

For the $\gamma$-entropy of the joint data distribution, Proposition 5 needs to be applied twice, first with respect to $p_{\text{data}}(x)$ then with respect to $q_{\phi,\nu}(z_1|x)$:

$$\mathcal{H}_\gamma(q_{\phi,\nu}) = -\left(\iiint q_{\phi,\nu}(x,z_1,z_2)^{1+\gamma}dz_2dz_1dx\right)^{\frac{1}{1+\gamma}}$$

$$= -\left(\iint\left(\int q_{\phi,\nu}(z_2|x,z_1)^{1+\gamma}dz_2\right)q_{\phi,\nu}(z_1|x)^{1+\gamma}p_{\text{data}}(x)^{1+\gamma}dz_1dx\right)^{\frac{1}{1+\gamma}}$$

$$= -\underbrace{C_{\nu+m_1+n,m_2}^{\frac{\gamma}{1+\gamma}}\left(1+\frac{m_1+n}{\nu}\right)^{\frac{\gamma m_2}{2(1+\gamma)}}\left(1+\frac{m_2}{\nu+m_1+n-2}\right)^{\frac{1}{1+\gamma}}}_{=:\widetilde{C}_1}$$

$$\times\left(\iint |\Sigma_\phi(x,z_1)|^{-\frac{\gamma}{2}}q_{\phi,\nu}(z_1|x)^{1+\gamma}p_{\text{data}}(x)^{1+\gamma}dz_1dx\right)^{\frac{1}{1+\gamma}}$$

$$\approx -\widetilde{C}_1 \left( \int \left( \int |\Sigma_\phi(x, z_1)|^{-\frac{\gamma}{2}} \, q_{\phi,\nu}(z_1|x)^{1+\gamma} dz_1 \right)^{\frac{1}{1+\gamma}} p_{\text{data}}(x)dx - H(p_{\text{data}}) \cdot \gamma \right)$$

$$\approx -\widetilde{C}_1 \int \left( \int |\Sigma_\phi(x, z_1)|^{-\frac{\gamma}{2(1+\gamma)}} \, q_{\phi,\nu}(z_1|x)dz_1 - H(q_{\phi,\nu}(\cdot|x)) \cdot \gamma \right) p_{\text{data}}(x)dx + \text{const.}$$

$$= -\widetilde{C}_1 \mathbb{E}_{x \sim p_{\text{data}}} \left\{ \mathbb{E}_{z_1 \sim q_{\phi,\nu}(\cdot|x)} \left[ |\Sigma_\phi(x, z_1)|^{-\frac{\gamma}{2(1+\gamma)}} \right] - \frac{\gamma}{2} \log |\Lambda_\phi(x)| \right\} + \text{const.}$$

Note that a new order $O(\gamma)$ correction term $\log |\Lambda_\phi(x)|$ appears in the inner approximation due to the dependency of the differential entropy of $q_{\phi,\nu}(\cdot|x)$ on $x$.

Next, we evaluate the required integrals in the $\gamma$-power cross-entropy,

$$\iiint p_{\theta,\nu}(x, z_1, z_2)^{1+\gamma} dx dz_2 dz_1 = \mathbb{E}_{z_1 \sim p_{Z,\nu}} \mathbb{E}_{z_2 \sim p_{\theta,\nu}(\cdot|z_1)} \mathbb{E}_{x \sim p_{\theta,\nu}(\cdot|z_1, z_2)} \left[ p_{\theta,\nu}(x, z_1, z_2)^\gamma \right]$$

$$= C_{\nu, m_1+m_2+n}^\gamma \sigma_z^{-\gamma m_2} \sigma_x^{-\gamma n} \mathbb{E}_{z_1 \sim p_{Z,\nu}} \mathbb{E}_{z_2 \sim p_{\theta,\nu}(\cdot|z_1)} \left[ 1 + \frac{\|z_1\|^2}{\nu} + \frac{\|z_2 - \zeta_\theta(z_1)\|^2}{\nu \sigma_z^2} \right.$$

$$+ \frac{1}{\nu \sigma_x^2} \text{tr} \left( \frac{\nu + m_1 + m_2}{\nu + m_1 + m_2 - 2} \cdot \frac{1 + \nu^{-1} \|z_1\|^2 + \nu^{-1} \sigma_z^{-2} \|z_2 - \zeta_\theta(z_1)\|^2}{1 + \nu^{-1}(m_1 + m_2)} \sigma_x^2 I \right) \right]$$

$$= C_{\nu, m_1+m_2+n}^\gamma \sigma_z^{-\gamma m_2} \sigma_x^{-\gamma n} \frac{\nu + m_1 + m_2 + n - 2}{\nu + m_1 + m_2 - 2}$$

$$\times \mathbb{E}_{z_1 \sim p_{Z,\nu}} \mathbb{E}_{z_2 \sim p_{\theta,\nu}(\cdot|z_1)} \left[ 1 + \frac{\|z_1\|^2}{\nu} + \frac{\|z_2 - \zeta_\theta(z_1)\|^2}{\nu \sigma_z^2} \right]$$

$$= C_{\nu, m_1+m_2+n}^\gamma \sigma_z^{-\gamma m_2} \sigma_x^{-\gamma n} \left( 1 + \frac{m_1 + m_2 + n}{\nu - 2} \right).$$

Furthermore,

$$\iiint q_{\phi,\nu}(x, z_1, z_2) p_{\theta,\nu}(x, z_1, z_2)^\gamma dz_2 dz_1 dx$$

$$= \mathbb{E}_{x \sim p_{\text{data}}} \mathbb{E}_{z_1 \sim q_{\phi,\nu}(\cdot|x)} \mathbb{E}_{z_2 \sim q_{\phi,\nu}(\cdot|x, z_1)} \left[ p_{\theta,\nu}(x, z_1, z_2)^\gamma \right]$$

$$= C_{\nu, m_1+m_2+n}^\gamma \sigma_z^{-\gamma m_2} \sigma_x^{-\gamma n}$$

$$\times \mathbb{E}_{x \sim p_{\text{data}}} \mathbb{E}_{z_1 \sim q_{\phi,\nu}(\cdot|x)} \mathbb{E}_{z_2 \sim q_{\phi,\nu}(\cdot|x, z_1)} \left[ 1 + \frac{\|z_1\|^2}{\nu} + \frac{\|z_2 - \zeta_\theta(z_1)\|^2}{\nu \sigma_z^2} + \frac{\|x - \mu_\theta(z_1, z_2)\|^2}{\nu \sigma_x^2} \right]$$

$$= C_{\nu, m_1+m_2+n}^\gamma \sigma_z^{-\gamma m_2} \sigma_x^{-\gamma n} \mathbb{E}_{x \sim p_{\text{data}}} \left\{ 1 + \frac{1}{\nu} \left( \|\zeta_\phi(x)\|^2 + \frac{\nu}{\nu + n - 2} \text{tr} \, \Lambda_\phi(x) \right) \right.$$

$$+ \frac{1}{\nu \sigma_z^2} \mathbb{E}_{z_1 \sim q_{\phi,\nu}(\cdot|x)} \left[ \|\mu_\phi(x, z_1) - \zeta_\theta(z_1)\|^2 + \frac{\nu}{\nu + m_1 + n - 2} \text{tr} \, \Sigma_\phi(x, z_1) \right]$$

$$+ \left. \frac{1}{\nu \sigma_x^2} \mathbb{E}_{z_1 \sim q_{\phi,\nu}(\cdot|x)} \mathbb{E}_{z_2 \sim q_{\phi,\nu}(\cdot|x, z_1)} \|x - \mu_\theta(z_1, z_2)\|^2 \right\}.$$

Therefore, the $\gamma$-power cross-entropy is obtained as

$$\mathcal{C}_\gamma(q_{\phi,\nu}, p_{\theta,\nu})$$

$$= \left( \iiint q_{\phi,\nu}(x, z_1, z_2) p_{\theta,\nu}(x, z_1, z_2)^\gamma dz_2 dz_1 dx \right) \left( \iiint p_{\theta,\nu}(x, z_1, z_2)^{1+\gamma} dx dz_2 dz_1 \right)^{-\frac{\gamma}{1+\gamma}}$$

$$= -\underbrace{\left( C_{\nu, m_1+m_2+n}^\gamma \sigma_z^{-\gamma m_2} \sigma_x^{-\gamma n} \right)^{\frac{1}{1+\gamma}} \left( 1 + \frac{m_1 + m_2 + n}{\nu - 2} \right)^{-\frac{\gamma}{1+\gamma}}}_{=: \widetilde{C}_2}$$

$$\times \mathbb{E}_{x \sim p_{\text{data}}} \left\{ 1 + \frac{1}{\nu} \left( \|\zeta_\phi(x)\|^2 + \frac{\nu}{\nu + n - 2} \text{tr} \, \Lambda_\phi(x) \right) \right.$$

$$+ \frac{1}{\nu\sigma_z^2}\mathbb{E}_{z_1 \sim q_{\phi,\nu}(\cdot|x)}\left[\|\mu_\phi(x,z_1) - \zeta_\theta(z_1)\|^2 + \frac{\nu}{\nu + m_1 + n - 2}\operatorname{tr}\Sigma_\phi(x,z_1)\right]$$

$$+ \frac{1}{\nu\sigma_x^2}\mathbb{E}_{z_1 \sim q_{\phi,\nu}(\cdot|x)}\mathbb{E}_{z_2 \sim q_{\phi,\nu}(\cdot|x,z_1)}\|x - \mu_\theta(z_1,z_2)\|^2\Bigg\}.$$

Putting everything together and tidying constants, we finally obtain the $\gamma$-loss for $t^3$HVAE,

$$\widetilde{\mathcal{L}}_\gamma(\theta, \phi) = \frac{1}{2}\,\mathbb{E}_{x \sim p_{\mathrm{data}}}\Bigg\{ \underbrace{\frac{1}{\sigma_x^2}\mathbb{E}_{z_1 \sim q_{\phi,\nu}(\cdot|x)}\mathbb{E}_{z_2 \sim q_{\phi,\nu}(\cdot|x,z_1)}\|x - \mu_\theta(z_1,z_2)\|^2}_{\text{reconstruction error}}$$

$$+ \underbrace{\|\zeta_\phi(x)\|^2 + \frac{\nu}{\nu + n - 2}\operatorname{tr}\Lambda_\phi(x) + \frac{\gamma\nu\widetilde{C}_1}{2\widetilde{C}_2}\log|\Lambda_\phi(x)|}_{\text{regularizer for } q_{\phi,\nu}(z_1|x)}$$

$$+ \frac{1}{\sigma_z^2}\mathbb{E}_{z_1 \sim q_{\phi,\nu}(\cdot|x)}\left[\underbrace{\|\mu_\phi(x,z_1) - \zeta_\theta(z_1)\|^2 + \frac{\nu \cdot \operatorname{tr}\Sigma_\phi(x,z_1)}{\nu + m_1 + n - 2} - \frac{\nu\sigma_z^2\widetilde{C}_1}{\widetilde{C}_2}|\Sigma_\phi(x,z_1)|^{-\frac{\gamma}{2(1+\gamma)}}}_{\text{regularizer for } q_{\phi,\nu}(z_2|x,z_1)}\right]\Bigg\}.$$

$$(23)$$

Similarly to the $\gamma$-loss, we see that $\widetilde{\mathcal{L}}_\gamma(\theta, \phi)$ consists of the reconstruction error and a sum of regularizing terms for each of the first and second level encoders. It is also possible to replace the term $\log|\Lambda_\phi(x)|$ with a constant times $-|\Lambda_\phi(x)|^{-\frac{\gamma}{2(1+\gamma)}}$ for consistency by converting the order $O(\gamma)$ correction term in Proposition 5 from differential entropy to $\gamma$-power entropy.

## C EXPERIMENTAL DETAILS

All experiments are implemented via Python 3.8.10 with the PyTorch package (Paszke et al., 2019) version 1.13.1+cu117, and run on Linux Ubuntu 20.04 with Intel® Xeon® Silver 4114 @ 2.20GHz processors, an Nvidia Titan V GPU with 12GB memory, CUDA 11.3 and cuDNN 8.2. The code is available on Github.

### C.1 REPARAMETRIZATION TRICK FOR $t^3$VAE

Recall that for the Gaussian VAE, given $N$ data points $x^{(1)}, \cdots, x^{(N)}$, a Monte Carlo estimate of the observed ELBO is computed using $L$ samples $z^{(i,1)}, \cdots, z^{(i,L)}$ drawn from $q_\phi(\cdot|x^{(i)})$,

$$\widehat{\mathcal{L}}(\theta, \phi) = \frac{1}{N}\sum_{i=1}^N\left[\frac{1}{L}\sum_{\ell=1}^L \log p_\theta(x^{(i)}|z^{(i,\ell)}) - \mathcal{D}_{\mathrm{KL}}(q_\phi(z|x^{(i)}) \,\|\, p_Z(z))\right].$$

In order to backpropagate gradients through stochastic nodes, the reparametrization trick must be used wherein $z^{(i,\ell)} = \mu_\phi(x^{(i)}) + \sigma_\phi(x^{(i)}) \odot \epsilon^{(i,\ell)}$ and $\epsilon^{(i,\ell)}$ is sampled from $\mathcal{N}_m(0, I)$. Since the t-distribution behaves similarly under affine transformations, this mechanism is simple to implement for our model.

A multivariate t-distribution $T \sim t_d(\mu, \Sigma, \nu)$ may be constructed from a multivariate centered Gaussian $Z \sim \mathcal{N}_d(0, \Sigma)$ and an independent chi-squared variable $V \sim \chi^2(\nu)$ via

$$T \overset{d}{=} \mu + \frac{Z}{\sqrt{V/\nu}}.$$

We use an encoder with diagonal covariance,

$$q_{\phi,\nu}(z|x) = t_m\left(z\,\bigg|\,\mu_\phi(x), \frac{1}{1 + \nu^{-1}n}\operatorname{diag}\sigma_\phi^2(x), \nu + n\right),$$

hence we may obtain $z^{(i,\ell)}$ by drawing $\epsilon^{(i,\ell)} \sim \mathcal{N}_m(0, I)$ and $\delta^{(i,\ell)} \sim \chi^2(\nu + n)$ independently and computing

$$
\begin{aligned}
z^{(i,\ell)} &= \mu_\phi(x^{(i)}) + \frac{1}{\sqrt{\delta^{(i,\ell)}/(\nu+n)}} \frac{\sigma_\phi(x^{(i)})}{\sqrt{1 + \nu^{-1}n}} \odot \epsilon^{(i,\ell)} \\
&= \mu_\phi(x^{(i)}) + \sqrt{\frac{\nu}{\delta^{(i,\ell)}}} \sigma_\phi(x^{(i)}) \odot \epsilon^{(i,\ell)}.
\end{aligned}
\tag{24}
$$

## C.2 DETAILS ON SYNTHETIC DATASETS

We first generate 200K train data, 200K validation data, and 500K test data from the heavy-tailed bimodal distribution (22). We then construct decoder and encoder networks with a common multi-layer perceptron architecture, which consists of two fully connected layers each with 64 nodes and Leaky ReLU activations. In the training process, we use a batch size of 128 and employ the Adam optimizer (Kingma & Ba, 2014) with a learning rate of $1 \times 10^{-3}$ and weight decay $1 \times 10^{-4}$. Moreover, we adapt early stopping using validation data with patience 15 to prevent overfitting. All models finished training within 80 epochs except VAE-st.

After completing training, we generate 500K samples from each model. The large sample size is necessary in order to accurately plot the far tails. For MMD hypothesis testing, we employ the fast linear time bootstrap test (Gretton et al., 2012) with 100K subsamples from each distribution and 1K repetitions. For the tail distributions, we employ the same test with appropriate subsampling to match sample sizes.

Moreover, we also compare the reconstruction loss of $t^3$VAE and Gaussian VAE in Table C1. This comparison is valid since both the ELBO (2) and $\gamma$-loss (20) optimize the same MSE loss with different regularizers. We observe that models with smaller $\nu$ exhibit smaller reconstruction loss, demonstrating the relaxation of regularization discussed in Section 3.2. Smaller losses can of course be achieved by $\beta$-VAE by decreasing $\beta$, which is close to a simple autoencoder.

The empty blocks for $\beta$-VAE ($\beta = 2.0$) and VAE-st ($\nu = 15.0$) are due to no tail samples being generated at all. Also, the reconstruction loss of Student-$t$ VAE cannot be numerically compared as it assumes learnable variance $\sigma^2$.

## C.3 DETAILS ON CELEBA AND CIFAR100-LT

The original CIFAR100 dataset is provided by Krizhevsky (2009). CIFAR100 consists of 60K $32 \times 32$ color images with 100 balanced classes, divided into 50K train images and 10K test images. The CelebA dataset is provided by Liu et al. (2015; 2018). The dataset consists of 202,599 face images from 10,177 celebrities, which are split into 182,637 training and 19,962 test images following the recommendation provided in the CelebA official documentation. Each image is subjected to uniform center-cropping, specifically by $148 \times 148$ pixels, followed by downsizing to $64 \times 64$ pixels.

All VAE models are trained for 50 epochs using a batch size of 128 and a latent variable dimension of 64 with the Adam optimizer. We tune learning rates separately for each model within the range of $5 \times 10^{-5} \sim 5 \times 10^{-4}$ and present the best result. In particular, good results are obtained when the learning rate is set to $8 \times 10^{-5} \sim 1 \times 10^{-4}$ in CelebA, and $2 \times 10^{-4} \sim 4 \times 10^{-4}$ in CIFAR100-LT. In the case of FactorVAE, a separate Adam optimizer is used for the discriminator with a learning rate of $1 \times 10^{-5}$.

The basic encoder and decoder architectures for most models are based on the public GitHub repository (Subramanian, 2020) with modifications. The encoder network consists of 5 convolutional layers with hidden layer sizes of 128, 256, 512, 1024, 2048. Batch normalization and ReLU activation are applied after each convolutional layer. Two fully connected layers for $\mu_\phi(x)$ and $\Sigma_\phi(x)$ then flatten the encoder output to the latent space. The decoder network is the reverse of the encoder architecture, with 5 transposed convolutional layers instead. The final part of the decoder is a pair of transposed convolutional layers and a tanh activation.

Hierarchical models, HVAE and $t^3$HVAE, maintain the encoder and decoder networks of the original model while adding a second encoder $q_{\phi,\nu}(z_2|x, z_1)$ and a layer that estimates the mean of the second prior $\zeta_\theta(z_1)$ using a multi-layer perceptron. The dimension of $z_2$ is set to half that of $z_1$, and skip

Table C1: Full results of MMD test $p$-values of 1-dimensional synthetic data analysis. Rejected values are shown in red.

| Model | Parameter | MMD test $p$-values | | | Recon loss |
|---|---|---|---|---|---|
| | | Full | Left tail | Right tail | |
| $t^3$VAE | $\nu = 9.0$ | < 0.001 | 0.130 | 0.580 | 0.707 |
| | $\nu = 12.0$ | 0.026 | 0.330 | 0.347 | 0.744 |
| | $\nu = 15.0$ | 0.008 | 0.363 | 0.622 | 0.831 |
| | $\nu = 18.0$ | 0.322 | 0.377 | 0.693 | 0.847 |
| | $\nu = 21.0$ | 0.372 | 0.188 | 0.838 | 0.853 |
| VAE | - | 0.514 | 0.036 | 0.003 | 1.028 |
| $\beta$-VAE | $\beta = 0.1$ | 0.614 | 0.011 | < 0.001 | 0.102 |
| | $\beta = 0.2$ | 0.137 | < 0.001 | 0.032 | 0.210 |
| | $\beta = 0.5$ | 0.050 | 0.007 | 0.027 | 0.506 |
| | $\beta = 2.0$ | < 0.001 | - | - | - |
| Student $t$-VAE | - | 0.587 | 0.291 | 0.114 | - |
| DE-VAE | $\nu = 9.0$ | 0.943 | 0.424 | 0.814 | 1.012 |
| | $\nu = 12.0$ | 0.068 | 0.222 | 0.406 | 1.018 |
| | $\nu = 15.0$ | 0.481 | 0.046 | 0.443 | 0.997 |
| | $\nu = 18.0$ | 0.763 | 0.219 | 0.411 | 0.984 |
| | $\nu = 21.0$ | 0.597 | 0.376 | 0.146 | 0.985 |
| VAE-st | $\nu = 9.0$ | < 0.001 | 0.179 | 0.460 | 1.062 |
| | $\nu = 12.0$ | < 0.001 | 0.953 | 0.643 | 1.057 |
| | $\nu = 15.0$ | < 0.001 | 0.363 | - | 1.045 |
| | $\nu = 18.0$ | < 0.001 | 0.925 | < 0.001 | 0.981 |
| | $\nu = 21.0$ | < 0.001 | 0.326 | < 0.001 | 1.094 |

Table C2: Full results of MMD test $p$-values of 2-dimensional synthetic data analysis. Rejected values are shown in red.

| Model | Parameter | MMD test $p$-values | | |
|---|---|---|---|---|
| | | Full | Left tail | Right tail |
| $t^3$VAE | $\nu = 30.0$ | 0.276 | 0.214 | 0.213 |
| | $\nu = 40.0$ | 0.716 | 0.130 | < 0.001 |
| | $\nu = 50.0$ | 0.766 | 0.142 | 0.004 |
| | $\nu = 60.0$ | 0.665 | 0.060 | 0.002 |
| | $\nu = 70.0$ | 0.773 | 0.100 | 0.002 |
| VAE | - | 0.116 | 0.004 | < 0.001 |
| $\beta$-VAE | $\beta = 0.1$ | 0.631 | < 0.001 | < 0.001 |
| | $\beta = 0.2$ | 0.359 | < 0.001 | < 0.001 |
| | $\beta = 0.5$ | 0.251 | < 0.001 | 0.015 |
| Student $t$-VAE | - | 0.530 | < 0.001 | < 0.001 |
| DE-VAE | $\nu = 3.0$ | 0.624 | < 0.002 | 0.057 |
| | $\nu = 5.0$ | 0.672 | < 0.001 | < 0.001 |
| | $\nu = 7.0$ | 0.452 | < 0.001 | < 0.001 |
| | $\nu = 9.0$ | 0.539 | < 0.001 | < 0.001 |
| VAE-st | $\nu = 3.0$ | 0.485 | < 0.001 | < 0.001 |
| | $\nu = 5.0$ | 0.092 | < 0.001 | 0.020 |
| | $\nu = 7.0$ | 0.250 | < 0.001 | < 0.001 |
| | $\nu = 9.0$ | 0.156 | < 0.001 | 0.020 |

connections are implemented by concatenating the connected input with the original input. After training, we load the best model of all epochs and evaluate FID scores for the whole dataset and each class separately.

**The complete version of Table 2.** In Tables C3 and C4, we provide the complete version of Table 2 including the FID scores with varying hyperparameters. We verify that $t^3$VAE achieves the lowest FID scores, regardless of $\nu$. In addition, we include the FID scores of Hierarchical model at the bottom of Table 2, emphasizing that $t^3$HVAE consistently yields more favorable outcomes compared to the conventional model.

Table C3: CelebA FID scores for overall and selected attribute classes with varying hyperparameters for each model, including Pale Skin (Pale) and Double Chin (Chin). We also include the Gaussian HVAE and $t^3$HVAE reconstruction scores.

| Framework | Parameter | Overall | Bald | Mst | Gray | Pale | Chin | No Bd | Young |
|---|---|---|---|---|---|---|---|---|---|
| $t^3$VAE | $\nu = 10$ | 39.4 | 66.5 | 61.5 | 67.2 | 54.6 | 57.2 | 40.1 | 46.7 |
| | $\nu = 5$ | 40.2 | 67.5 | 62.6 | 68.8 | 55.1 | 58.1 | 40.8 | 47.8 |
| | $\nu = 2.5$ | 40.0 | 67.6 | 62.0 | 68.0 | 55.8 | 57.7 | 40.7 | 47.2 |
| | $\nu = 2.1$ | 40.4 | 68.5 | 62.5 | 68.0 | 55.8 | 58.4 | 41.0 | 47.6 |
| VAE | $\kappa = 1$ | 57.9 | 85.8 | 79.7 | 91.0 | 70.8 | 78.2 | 58.4 | 69.1 |
| | $\kappa = 1.5$ | 73.2 | 105.3 | 96.4 | 114.5 | 85.3 | 97.5 | 73.8 | 84.1 |
| $\beta$-VAE | $\beta = 0.5$ | 50.9 | 81.0 | 72.4 | 84.3 | 66.0 | 71.6 | 51.4 | 60.1 |
| | $\beta = 0.25$ | 46.7 | 76.5 | 69.3 | 79.1 | 60.7 | 66.9 | 46.9 | 55.7 |
| | $\beta = 0.1$ | 42.1 | 72.1 | 65.3 | 74.8 | 56.1 | 62.4 | 42.5 | 51.3 |
| | $\beta = 0.05$ | 40.4 | 69.3 | 62.7 | 71.1 | 55.1 | 59.8 | 40.9 | 49.0 |
| Student-$t$ VAE | - | 78.4 | 112.0 | 104.2 | 118.7 | 91.7 | 100.7 | 78.6 | 88.9 |
| DE-VAE | $\nu = 10$ | 59.7 | 89.2 | 83.8 | 95.6 | 73.8 | 81.4 | 59.9 | 69.4 |
| | $\nu = 5$ | 58.9 | 89.6 | 84.3 | 94.9 | 72.8 | 80.2 | 59.1 | 68.9 |
| | $\nu = 2.5$ | 59.0 | 84.6 | 81.3 | 93.1 | 72.1 | 78.2 | 59.4 | 68.1 |
| | $\nu = 2.1$ | 59.1 | 90.5 | 83.7 | 94.5 | 72.9 | 80.9 | 59.4 | 69.1 |
| Tilted VAE | $\tau = 20$ | 49.5 | 79.8 | 73.3 | 82.1 | 63.8 | 70.3 | 49.9 | 59.3 |
| | $\tau = 30$ | 45.9 | 75.1 | 69.5 | 77.1 | 60.6 | 65.4 | 46.3 | 54.7 |
| | $\tau = 40$ | 45.0 | 74.4 | 68.1 | 76.6 | 59.9 | 65.1 | 45.2 | 54.1 |
| | $\tau = 50$ | 42.6 | 73.0 | 65.4 | 73.7 | 57.8 | 62.1 | 42.9 | 50.8 |
| FactorVAE | $\gamma_{tc} = 20$ | 61.2 | 96.7 | 88.7 | 100.1 | 77.1 | 84.8 | 61.3 | 71.8 |
| | $\gamma_{tc} = 10$ | 60.8 | 95.4 | 87.2 | 101.2 | 75.6 | 85.5 | 60.8 | 72.4 |
| | $\gamma_{tc} = 5$ | 59.8 | 91.7 | 85.7 | 95.2 | 74.2 | 81.5 | 59.9 | 70.0 |
| | $\gamma_{tc} = 2.5$ | 60.8 | 92.5 | 86.6 | 96.8 | 73.6 | 82.5 | 60.8 | 71.1 |
| HVAE | - | 56.9 | 85.4 | 78.9 | 88.1 | 68.0 | 89.1 | 56.7 | 67.2 |
| $t^3$HVAE | $\nu = 10$ | 37.5 | 65.9 | 59.6 | 67.7 | 52.8 | 56.1 | 37.8 | 45.2 |
| | $\nu = 5$ | 37.0 | 65.4 | 59.3 | 67.2 | 52.6 | 55.9 | 37.4 | 44.5 |
| | $\nu = 2.5$ | 36.7 | 64.6 | 58.6 | 66.1 | 52.5 | 54.9 | 37.1 | 43.8 |

**Comparing $\beta$-VAE and $t^3$VAE.** For more accurate comparisons between $\beta$-VAE and $t^3$VAE in CelebA generation, we vary $\beta$ within a range from 0.001 to 1. We again note that the performance of $t^3$VAE is not sensitive to $\nu$, so no fine-tuning is required. Figure C2 shows that $\beta$-VAE performs best when $\beta$ is set to 0.05 and becomes less effective as $\beta$ deviates further from 0.05.

This aspect is also considered in the reconstruction experiment design. If $\beta$ is set to ever smaller values, the reconstruction results may appear to improve over other models. However, this is because $\beta$-VAE essentially regresses to a raw autoencoder as $\beta \to 0$, so that while reconstruction performance can be improved, generation performance will simultaneously deteriorate. We therefore do not tune for extremely small values of $\beta$.

Table C4: FID scores (Total data and lower 10% of tailed class) on CIFAR100-LT with varying imbalance factor $\rho$.

| Framework | Parameter | $\rho = 1$ | | $\rho = 10$ | | $\rho = 50$ | | $\rho = 100$ | |
|---|---|---|---|---|---|---|---|---|---|
| | | Total | Rare | Total | Rare | Total | Rare | Total | Rare |
| $t^3$VAE | $\nu = 2.1$ | 100.0 | 142.2 | 101.2 | 146.2 | 123.5 | 166.2 | 130.2 | 170.8 |
| | $\nu = 2.5$ | 98.5 | 140.9 | 100.4 | 144.8 | 128.5 | 170.7 | 130.0 | 172.1 |
| | $\nu = 5$ | 100.2 | 141.8 | 103.8 | 147.7 | 126.8 | 171.3 | 131.0 | 173.7 |
| | $\nu = 10$ | 97.5 | 140.9 | 102.8 | 147.6 | 128.5 | 170.8 | 128.7 | 170.9 |
| VAE | $\kappa = 1$ | 256.1 | 305.7 | 267.2 | 315.5 | 277.4 | 332.7 | 287.3 | 342.3 |
| | $\kappa = 1.5$ | 274.2 | 298.9 | 290.5 | 313.1 | 296.7 | 318.9 | 297.7 | 319.6 |
| $\beta$-VAE | $\beta = 0.5$ | 179.5 | 210.2 | 198.2 | 230.8 | 214.6 | 245.8 | 223.7 | 252.9 |
| | $\beta = 0.25$ | 142.1 | 178.0 | 160.0 | 197.7 | 169.2 | 204.5 | 182.5 | 217.5 |
| | $\beta = 0.1$ | 114.0 | 154.7 | 130.4 | 170.8 | 138.5 | 178.4 | 160.6 | 199.0 |
| Student-$t$ VAE | – | 259.5 | 315.8 | 314.1 | 353.1 | 323.7 | 364.1 | 333.4 | 374.3 |
| DE-VAE | $\nu = 2.1$ | 230.4 | 258.5 | 250.0 | 280.0 | 255.0 | 282.7 | 254.4 | 285.6 |
| | $\nu = 2.5$ | 219.4 | 250.0 | 250.2 | 281.6 | 256.7 | 285.1 | 258.5 | 303.2 |
| | $\nu = 5$ | 232.6 | 260.9 | 252.9 | 284.6 | 252.0 | 281.3 | 269.7 | 306.7 |
| | $\nu = 10$ | 230.9 | 262.0 | 250.5 | 278.0 | 258.0 | 283.6 | 272.4 | 303.5 |
| Tilted VAE | $\tau = 20$ | 113.8 | 154.1 | 131.2 | 172.6 | 149.6 | 189.5 | 181.8 | 221.8 |
| | $\tau = 30$ | 104.4 | 146.1 | 124.1 | 166.5 | 143.3 | 184.5 | 179.1 | 220.6 |
| | $\tau = 40$ | 101.0 | 142.5 | 123.9 | 167.4 | 179.1 | 220.6 | 172.4 | 223.6 |
| | $\tau = 50$ | 101.0 | 142.8 | 126.1 | 168.4 | 147.0 | 187.8 | 193.2 | 229.7 |
| FactorVAE | $\gamma_{tc} = 20$ | 238.8 | 266.2 | 277.6 | 322.8 | 273.8 | 301.8 | 269.1 | 297.8 |
| | $\gamma_{tc} = 10$ | 240.4 | 268.6 | 273.5 | 324.3 | 270.5 | 298.1 | 270.2 | 298.2 |
| | $\gamma_{tc} = 5$ | 232.3 | 263.3 | 272.5 | 323.6 | 275.6 | 306.3 | 270.1 | 296.7 |
| | $\gamma_{tc} = 2.5$ | 236.0 | 264.4 | 275.7 | 328.9 | 269.2 | 298.2 | 269.8 | 297.8 |

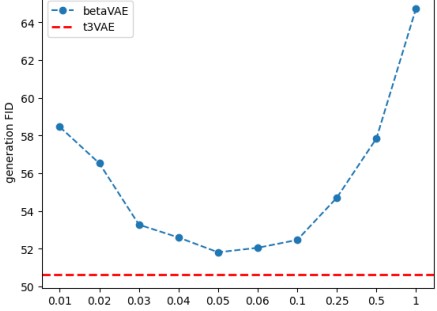

Figure C2: Generation FID scores against $\beta$ values for $\beta$-VAE.

| Framework | Parameter | FID | Framework | Parameter | FID |
|---|---|---|---|---|---|
| $t^3$VAE | $\nu = 10$ | 50.6 | DE-VAE | $\nu = 10$ | 62.8 |
| | $\nu = 5$ | 50.9 | | $\nu = 5$ | 61.6 |
| | $\nu = 2.5$ | 50.6 | | $\nu = 2.5$ | 58.9 |
| | $\nu = 2.1$ | 50.7 | | $\nu = 2.1$ | 60.4 |
| $\beta$-VAE | $\beta = 0.5$ | 57.8 | Tilted VAE | $\tau = 20$ | 58.4 |
| | $\beta = 0.25$ | 54.7 | | $\tau = 30$ | 59.2 |
| | $\beta = 0.1$ | 52.4 | | $\tau = 40$ | 60.8 |
| | $\beta = 0.05$ | 51.8 | | $\tau = 50$ | 61.1 |
| VAE | $\kappa = 1$ | 79.6 | FactorVAE | $\gamma_{tc} = 2.5$ | 67.1 |
| | $\kappa = 1.5$ | 64.7 | | $\gamma_{tc} = 5$ | 68.0 |
| Student-$t$ VAE | – | 82.3 | | $\gamma_{tc} = 10$ | 73.1 |
| | | | | $\gamma_{tc} = 20$ | 78.6 |

Table C5: CelebA FID scores for overall classes with varying hyperparameters for each model.

