# OpenReview forum: "$t^3$-Variational Autoencoder: Learning Heavy-tailed Data with Student's t and Power Divergence"
_ICLR.cc/2024/Conference — ICLR 2024 poster_

### Official Review · Reviewer_1fkZ · 2023-10-31

**Soundness:** 3 good
**Presentation:** 3 good
**Contribution:** 3 good
**Rating:** 8
**Confidence:** 4

**Summary:**

In this paper, the authors introduce a novel VAE-like generative model ($t^3$VAE) in which the underlying distributions (for the prior, encoder, and decoder) are assumed to be (multivariate) Student's t. This idea leads to a better approximation of heavy-tailed densities, which is confirmed by experiments on both synthetic and real-world datasets. The presented (solid) theoretical justification stems from viewing classical VAE as a joint minimization process between two statistical manifolds and relies on replacing the KL divergence with the $\gamma$-power divergence.

**Strengths:**

(1) The idea behind $t^3$VAE (although somewhat natural) seems original and significant since it allows us to overcome (to some extent) the limitations of classical VAE.

(2) The proposed solution has (as the authors show) a solid theoretical background.

(3) The experimental results prove the superiority of $t^3$VAE over the state-of-the-art.

**Weaknesses:**

(1) The authors claim that their idea is extendable to hierarchical models. Although there is a theoretical justification for this in the appendix, the paper would benefit from corresponding experimental studies.

(2) Minor comments:

p. 2, l. 10 from bottom: the authors probably wanted to write "likelihood" (instead of "log-likelihood"),

p. 7, Tab. 3: $t^3$VAE $\to$ $t^3$VAE ($\nu=10$),

p. 9, l. 12: I suggest not to use the phrase "scores highest" (as in the case of FID "lower is better").

**Questions:**

(1) Have you considered providing experimental results for hierarchical architectures?

---

> ### Author Response · Authors · 2023-11-22
>
> We are extremely grateful for your positive and through review of our work! We have updated the paper following your suggestions.
>
> **Addressing Weaknesses & Questions**
>
> (1) The authors claim that their idea is extendable to hierarchical models. Although there is a theoretical justification for this in the appendix, the paper would benefit from corresponding experimental studies.
>
> We have newly implemented the two-layer hierarchical model ($t^3$HVAE) and compared against the Gaussian HVAE on double resolution images. The results are presented in Section 4.2, Figure 5 and Table C3. We demonstrate that the increased hierarchical depth allows $t^3$HVAE to learn more sophisticated images with substantially higher clarity and sharper detail compared to Gaussian HVAE, further justifying the generality and effectiveness of our theoretical framework.
>
> (2) Minor comments: Thank you for going through our paper in detail! We have edited the mistakes and generally improved the presentation of the text. Also, please see our responses to the other reviewers for some other points that we addressed.

---

> > ### Comment · Reviewer_1fkZ · 2023-11-22
> > **Thank you for your comment**
> >
> > I am satisfied with the authors' rebuttal. I keep my high rating of the contribution.

---

### Official Review · Reviewer_5yMY · 2023-11-01

**Soundness:** 3 good
**Presentation:** 3 good
**Contribution:** 3 good
**Rating:** 6
**Confidence:** 4

**Summary:**

The authors address the limitation of the standard Variational Autoencoder (VAE) that employs a Gaussian prior which sometimes fails to capture the intricate structures in data due to its fast-decaying tails. They propose a novel framework by leveraging the heavy-tailed nature of the Student's t-distributions for the prior, encoder, and decoder. They argue that this new formulation can better fit real-world datasets.  Empirical results suggest that t3VAE performs better in modeling low-density regions, and yields superior performance on the CelebA and imbalanced CIFAR-100 datasets.

**Strengths:**

- The use of Student's t-distributions as an alternative to the Gaussian prior in VAEs seem a fresh perspective. The heavy-tailed nature can indeed address some of the shortcomings of the standard Gaussian prior.

- The modification of the evidence lower bound to incorporate γ-power divergence might seem appropriate given the nature of the power families.

- Demonstrated superior performance on benchmark datasets, particularly on imbalanced/ longtailed version of CIFAR-100, which is a testament to the proposed model's robustness.

**Weaknesses:**

- The transition from Eq(5) to Eq(19) is not straightforward for me. While intuitively, adopting the γ-power divergence to the KL-divergence might seem appropriate, a central question arises: Does the γ-power divergence between the two manifolds still lead to the ELBO? This needs to be addressed for better clarity.

- Given that both t3VAE and the Student-t VAE integrate the Student's t-distribution into the VAE framework, a more granular comparison in the related work section would be enlightening. Highlighting the distinct features and advantages of t3VAE over the Student-t VAE would give readers a clearer understanding of its contributions.

-  I noticed that the hierarchical variants of the model weren't evaluated in the experimental section. Such an evaluation might provide insights into the model's performance in different configurations. I'd be keen to see these results.

- Incorporating the Student's t-distributions inherently increases the model's complexity. This can potentially introduce challenges in training and optimization. I recommend the authors delve into these potential challenges, discussing potential remedies and considerations for practitioners aiming to implement t3VAE.

**Questions:**

See Weakness.

---

> ### Author Response · Authors · 2023-11-22
>
> Thank you for your through review and suggestions, and raising a number of important issues! We have updated the manuscript to reflect our discussion, add experiments and improve presentation. Below are our responses to weaknesses which we hope can dispel any remaining concerns.
>
> **Addressing Weaknesses**
>
> * Does the $\gamma$-power divergence between the two manifolds still lead to the ELBO?
>
> Thank you for pointing out the issue. We agree this is an important distinction that needs to be clarified, and the following discussion has been added to the manuscript in Section 3.2.
>
> Due to the different expression for $\gamma$-power divergence and also the approximation step, the $\gamma$-loss does not function as a bound for log-likelihood, although joint minimization will still generally try to fit $p_\theta(x)$ to the true data distribution, maximizing likelihood while also keeping posteriors close. Its precise effect is a balance between reconstruction and regularization, which is explained throughout Section 3.3.
>
> We emphasize that the central philosophy of our framework is viewing the ELBO *not* as a lower bound of likelihood (which naturally leads to modifying the KL regularizer term as most VAE variants do), but as a divergence between joint distributions (which naturally leads to modifying the entire divergence as in Eq.19). The $\gamma$-power or log-$\gamma$ divergences are reasonable choices due to their information geometric properties and viability of closed-form computations with respect to t-distributions.
>
> But why does this work? Our experiments demonstrated that our framework is very effective even though we are not explicitly optimizing for likelihood. This can be explained in part by [1] which shows that $\gamma$-type divergence minimizers are M-estimators and proves asymptotic efficiency and robustness under heavy contamination, lending support to our approach as a solid alternative to MLE methods for statistical inference. Furthermore, for high-dimensional datasets $\gamma$ is quite small and the effect of changing divergence is simply less important compared to the effect of the coupled t-distributions (in particular the prior) whose degrees of freedom $\nu$ is small.
>
> * Given that both t3VAE and the Student-t VAE integrate the Student's t-distribution into the VAE framework, a more granular comparison in the related work section would be enlightening.
>
> We felt that the Student-t VAE design is less suitable to highlight in the introduction for a couple of reasons. The model uses a separate univariate t-distribution for each dimension (there is some confusion regarding dimension in the paper as well), and experiments only focus on overcoming the training instability issue for low-dimensional datasets. Moreover, performance improvement on image datasets in our experiments was not as significant compared to other models such as $\beta$-VAE and Tilted VAE. Similarly, DE-VAE also showed relatively poor reconstuction quality and VAE-st suffered from a range of inconsistencies and implementation issues. Hence we aimed for a broad overview of all the t-based VAEs rather than focusing on one single benchmark model.
>
> * I noticed that the hierarchical variants of the model weren't evaluated in the experimental section. Such an evaluation might provide insights into the model's performance in different configurations.
>
> We have newly implemented the two-layer hierarchical model ($t^3$HVAE) and compared against the Gaussian HVAE on double resolution images. The results are presented in Section 4.2, Figure 5 and Table C3. We demonstrate that the increased hierarchical depth allows $t^3$HVAE to learn more sophisticated images with substantially higher clarity and sharper detail compared to Gaussian HVAE, further justifying the generality and effectiveness of our theoretical framework.
>
> * Incorporating the Student's t-distributions inherently increases the model's complexity. This can potentially introduce challenges in training and optimization.
>
> Unlike models such as DE-VAE or VAE-st which require numerical integration to calculate the KL divergence between t-distributions for every data point, the explicit form of the $\gamma$-loss does not create any significant computational bottlenecks or new instability issues (e.g. division by zero variance, exploding gradients) compared to the original ELBO of Gaussian VAE. The same holds for the t-based reparametrization trick or multivariate t-distribution sampling process as detailed in Appendix C.1, requiring very minor additional computation. We also observed virtually equal runtimes in practice. Nonetheless we agree that this discussion is needed in the paper, and have merged into the final paragraph in Section 4 to give a general analysis on training cost and hyperparameter selection.
>
> [1] H. Fujisawa, S. Eguchi, 2008. Robust parameter estimation with a small bias against heavy contamination. J. Multivar. Anal. 99(9).

---

> > ### Comment · Reviewer_5yMY · 2023-11-23
> > **Thanks**
> >
> > Thanks to the authors for addressing my concerns.   I decided to keep my positive ratings.

---

### Official Review · Reviewer_fEQe · 2023-11-01

**Soundness:** 4 excellent
**Presentation:** 3 good
**Contribution:** 3 good
**Rating:** 8
**Confidence:** 3

**Summary:**

The authors develop a new variational autoencoder designed for heavy-tailed data by changing the prior, encoder, and decoder from Gaussian distributions to t-distributions, and the KL divergence to power divergence. The authors draw upon the EM perspective of VAEs developed by Han et al., 2021 to formulate a joint minimization objective, and use ideas from information geometry to tie together the exponent in the power divergence with the degrees of freedom in the t-distributions. The degrees of freedom hyperparameter is shown to affect the degree of regularization in the model. The model formulation is also described in terms of a Bayesian point of view. Four experiments are conducted comparing the performance of the proposed model with a selection of other competing VAEs: the first involving a univariate synthetic dataset (reporting histograms and MMD test p-values), the second involving a bivariate synthetic dataset (reporting MMD test p-values), the third involving the CelebA image dataset (reporting FID scores), and the fourth involving the CIFAR100-LT image dataset (reporting FID scores). In all cases, the model is shown to vastly outperform competing methods. All arguments are supported with ample theoretical derivations in the supplementary material.

**Strengths:**

- An interesting and important extension on the VAE framework to meaningfully deal with heavy-tailed data.
- A broad investigation and discussion into the underlying concepts, providing an excellent derivation of the method, and a good amount of detail extending into the appendices.
- Hyperparameters and model choices are thought out and well-justified. The authors could have simply attached a few of the underlying ideas together without much thought, but have chosen to go the extra mile.
- Model is shown to be highly effective compared to competitors, even on reasonably challenging image datasets where VAEs do not typically do well.
- Very well-written paper; no grammatical issues or typos that I could detect.

**Weaknesses:**

- Reporting p-values is not exactly ideal, especially when metrics are available.
- No general summary to assist with implementation.
- No examples conducted on datasets where heavy tails are known to be especially relevant (e.g. economic datasets).
- No discussion of, or comparisons to, similar developments in the normalizing flow literature (e.g. [1] and [2]).

[1] Jaini, P., Kobyzev, I., Yu, Y., & Brubaker, M. (2020). Tails of Lipschitz triangular flows. In International Conference on Machine Learning (pp. 4673-4681). PMLR.

[2] Liang, F., Mahoney, M., & Hodgkinson, L. (2022). Fat–Tailed Variational Inference with Anisotropic Tail Adaptive Flows. In International Conference on Machine Learning (pp. 13257-13270). PMLR.

**Questions:**

- Is the gamma-power divergence not the same as the Renyi divergence (up to constants)? How do they differ?
- Can you provide an algorithm environment to show how this should be implemented at a glance? I appreciate the presentation and the derivation of the model, but any reader looking to quickly implement it is likely to have trouble if they do not thoroughly read the paper.
- What do the MMD values themselves look like? These might be more useful to report than the p-values.
- Since you know the densities explicitly in the synthetic tests, you could use KSD instead of MMD, as this will have much improved statistical power. Have you tried this?
- Have you tried the model on any real datasets other than image sets?
- Do you know how the model compares with other normalizing flow models incorporating t-distributions?

---

> ### Author Response · Authors · 2023-11-22
>
> Thank you for your positive review and very through advice which helped us greatly to improve our paper! We also report that we have additionally implemented the hierarchical $t^3$HVAE and added experimental results. Below are our responses to questions (merged with weaknesses).
>
> * Is the gamma-power divergence not the same as the Renyi divergence (up to constants)? How do they differ?
>
> In fact, the following four divergences have similar definitions.
>
> $\alpha$-divergence $\frac{1}{\alpha(\alpha-1)}\left(\int p^\alpha q^{1-\alpha}-1\right)$
>
> Renyi divergence $\frac{1}{\alpha-1}\log\int p^\alpha q^{1-\alpha}$
>
> $\gamma$-power divergence $-\int p\left(\frac{q}{\lVert q\rVert_{1+\gamma}}\right)^\gamma +\lVert p\rVert_{1+\gamma}$
>
> log-$\gamma$ divergence $\frac{1}{\gamma(\gamma+1)}\log\int p^{\gamma+1} - \frac{1}{\gamma}\log\int pq^\gamma+\frac{1}{\gamma+1}\log\int q^{\gamma+1}$
>
> Despite the apparent similarities, they each possess rich geometric properties. The $\alpha$-geometry was originally studied by Amari and has analogous properties to KL divergence on the space of positive measures (being the intersection of the $f$- and Bregman families) but has a more complex dual connection when restricted to probability measures. The $\gamma$-geometry, studied by Eguchi, has a simple flatness structure where exponential families are replaced by power families and mixture geodesics are the same. Also, from a statistical perspective, $\alpha$ tunes between mass-covering or mode-seeking behavior, while $\gamma$ tunes between robustness and outlier-focused behavior [1].
>
> In the initial stages of our research, we considered all the above as well as robust $\beta$ and other $f$-divergences as replacements for KL, but decided upon $\gamma$-power divergence due to flatness properties and viability of closed-form computations. The $\alpha$ and $\beta$ type divergences have also been studied for variational inference in previous works [2,3] (although not with our joint optimization framework), while $\gamma$-divergence methods are quite new.
>
> * Can you provide an algorithm environment to show how this should be implemented at a glance?
>
> Thank you for the helpful suggestion! We have added an implementation summary of our framework as Algorithm 1 in the main text.
>
> * What do the MMD values themselves look like? These might be more useful to report than the p-values.
>
> When the RBF kernel $k(x,x') = \exp\left(-\frac{(x-x')^2}{2\sigma^2}\right)$ is implemented for the MMD bootstrap test, the hyperparameter $\sigma^2$ is automatically chosen as the median of $\{(x_i - x_j')^2\}_{i,j}$. In practice, $\sigma^2$ fluctuates between around 3 to 5. In this case, directly comparing MMD values may not be meaningful. The MMD values are also somewhat randomly distributed around -0.01~0.01 and not particularly informative on their own. Hence a bootstrap test must be run to obtain meaningful statistics, which is why we only report $p$-values.
>
> * Since you know the densities explicitly in the synthetic tests, you could use KSD instead of MMD, as this will have much improved statistical power. Have you tried this?
>
> Due to increased power and reduction of reliable samples due to tail truncation, KSD tends to reject *all* models including $t^3$VAE. Of course, no model can exactly approximate the true distribution even with infinitely many samples, which will lead to rejection for any sufficiently powerful test. Instead our goal was to be able to clearly delineate *relative* tail generation performance of $t^3$VAE and other models, which is well achieved by MMD testing: we proved that $t^3$VAE is harder to distinguish from test data compared to other models. We also considered various other testing methods, but ultimately chose MMD due to applicability to 2 dimensions and simplicity & speed of implementation.
>
> (continued below)

---

> > ### Author Response · Authors · 2023-11-22
> >
> > (continued)
> >
> > * Have you tried the model on any real datasets other than image sets?
> >
> > Due to time constraints, we had to focus our efforts into implementing the hierarchical $t^3$VAE to learn very high dimensional images (experiments added to Section 4.2, Figure 5 and Table C3). We note that many non-image datasets that are considered heavy-tailed (economic, weather, etc) are low dimensional, while $t^3$VAE exhibits the best performance improvements over models such as Student-t VAE in higher dimensions. Nonetheless, we will continue to experiment on other datasets for the camera-ready version.
> >
> > * Do you know how the model compares with other normalizing flow models incorporating t-distributions?
> >
> > Works such as [4,5,6] implement heavy-tailed base distributions for normalizing flows, which is conceptually similar to using a heavy-tailed prior for VAEs. Of course in principle, any data distribution can in principle be approximated arbitrarily closely if the flow/decoder networks are powerful enough. However, these papers argue intuitively and empirically that t-distributions ensure better robustness and generalization, and we believe the same arguments carry over to our model.
> >
> > We also take a step further (compared to prior-based alternatives) by also using t-distributions for the decoder and encoder, which enforces the power form Eq.(15) on the joint model distribution. If the decoder network $\mu_\theta$ is Lipschitz, this will explicitly ensure polynomial decay w.r.t. $x,z$. In fact, this control 'further down the line' seems to be key to $t^3$VAE's success as evidenced by our experiments against Student-t VAE, DE-VAE and VAE-st. This discussion has been added to the Related Works section.
> >
> > Moreover, this raises an intriguing question post factum: can the tails of distributions generated by normalizing flows also be more strictly controlled down the line (not just modifying the base distributions) to improve performance? We leave this as an interesting direction for future work.
> >
> > [1] Regli, Silva. Alpha-beta divergence for variational inference. 2018.
> >
> > [2] Li, Turner. Renyi divergence variational inference. NIPS 2016.
> >
> > [3] Akrami et al. Robust variational autoencoder. NeurIPS 2019.
> >
> > [4] Alexanderson, Henter. Robust model training and generalisation with Studentising flows. ICML 2020 workshop.
> >
> > [5] Amiri et al. Generating heavy-tailed synthetic data with normalizing flows. UAI 2020 workshop.
> >
> > [6] Laszkiewicz et al. Marginal tail-adaptive normalizing flows. ICML 2022.

---

### Meta-Review · Area_Chair_QTxX · 2023-12-09

**Metareview:**

This paper extends the standard VAE approach for Gaussian likelihoods (G^3VAE) Gaussian latent prior, Gaussian encoder and Gaussian decoder to t^3VAE where t=the t-distribution.

All the reviewers liked the paper, were satisfied with the discussion and are in favor of acceptance.

**Justification For Why Not Higher Score:**

The set-up seems very taliored to the specific Gaussian likelihood. It could have been useful with a more generic approach also valid for non-Gaussian likelihood and hierarchical models.

**Justification For Why Not Lower Score:**

The paper is a meaningful new contribution to the time-honoured VAE model.

---

### Decision · Program_Chairs · 2024-01-16

Accept (poster)